

# 1 The ABoVE L-band and P-band Airborne SAR Surveys

Charles E. Miller[1], Peter C. Griffith[2], Elizabeth Hoy[3], Naiara S. Pinto[1], Yunling Lou[1], Scott Hensley[1],
Bruce D. Chapman[1], Jennifer Baltzer[4], Kazem Bakian-Dogaheh[5], W. Robert Bolton[6], Laura Bourgeau-
Chavez[7], Richard H. Chen[1], Byung-Hun Choe[8], Leah Clayton[9], Thomas A. Douglas[10], Nancy French[7],
Jean E. Holloway[11], Gang Hong[8], Lingcao Huang[12], Go Iwahana[6], Liza Jenkins[7], John S. Kimball[13],
Tatiana Loboda[14], Michelle Mack[15], Philip Marsh[16], Roger J. Michaelides[17], Mahta Moghaddam[5],
Andrew Parsekian[18], Kevin Schaefer[12], Paul R. Siqueira[19], Debjani Singh[20], Alireza Tabatabaeenejad,[5]
Merritt Turetsky[21], Ridha Touzi,[8] Elizabeth Wig[22], Cathy J. Wilson[23], Paul Wilson[8], Stan D.
Wullschleger[20], Yonghong Yi[1,24], Howard A. Zebker[22], Yu Zhang[8], Yuhuan Zhao[5], Scott J. Goetz[25]
[1]Jet Propulsion Laboratory, California Institute of Technology, Pasadena, CA, 91109, USA
[2]NASA Goddard Space Flight Center / SSAI, Greenbelt, Maryland, USA
[3]NASA Goddard Space Flight Center / GST, Inc., Greenbelt, Maryland, USA
[4]Department of Biology, Wilfrid Laurier University, Waterloo, Ontario, Canada
[5]Department of Electrical Engineering, University of Southern California, Los Angeles, CA, USA
[6]International Arctic Research Center, University of Alaska Fairbanks, Fairbanks, AK 99775-7340 USA
[7]Michigan Tech Research Institute, Michigan Technological University, Ann Arbor, MI 48105 USA
[8]Canada Centre for Remote Sensing, Ottawa, Ontario K1A0E4, Canada
[9]Department of Earth & Planetary Sciences, Yale University, New Haven, CT, USA
[10]U.S. Army Cold Regions Research and Engineering Laboratory, Fort Wainwright, Alaska 99709 USA
[11]Department of Geography, University of Ottawa, Ottawa ON K1N 6N5, Canada
[12]National Snow and Ice Data Center (NSIDC), University of Colorado, Boulder, Colorado 80309-0449 USA
[13]NTSG, WA Franke College of Forestry & Conservation, The University of Montana, Missoula, Montana USA
[14]Department of Geographical Sciences, University of Maryland, College Park, Maryland 20742, USA
[15]Center for Ecosystem Science and Society and Department of Biological Sciences, Northern Arizona University, Flagstaff,
AZ 86011 USA
[16]Department of Geography and Cold Regions Research Centre, Wilfrid Laurier University, Waterloo, Ontario, Canada
[17]Department of Geophysics, Colorado School of Mines, Golden, CO, USA
[18]Department of Geology & Geophysics, University of Wyoming, 1000 E University Ave., Laramie, WY, USA
[19]Department of Electrical and Computer Engineering, University of Massachusetts, Amherst, MA 01003-9284 USA
[20]Biological and Environmental Systems Science Directorate, Oak Ridge National Laboratory, Oak Ridge, Tennessee USA
[21]Ecology and Evolutionary Biology Department, University of Colorado Boulder
[22]Department of Geophysics, Stanford University, Stanford, CA, USA
[23]Earth and Environmental Sciences Division, Los Alamos National Laboratory, Los Alamos, New Mexico USA
[24]Joint Institute for Regional Earth System Science & Engineering, The University of California, Los Angeles, CA 90095-
7228 USA
[25]School of Informatics, Computing, and Cyber Systems, Northern Arizona University, Flagstaff, AZ 86011 USA

*Correspondence to*: Charles E. Miller (charles.e.miller@jpl.nasa.gov)



**Abstract**

Permafrost-affected ecosystems of the Arctic-boreal zone in northwestern North America are undergoing profound transformation due to rapid climate change. NASA's Arctic Boreal Vulnerability Experiment (ABoVE) is investigating characteristics that make these ecosystems vulnerable or resilient to this change. ABoVE employs airborne synthetic aperture radar (SAR) as a powerful tool to characterize tundra, taiga, peatlands, and fens. Here, we present an annotated guide to the L-band and P-band airborne SAR data acquired during the 2017, 2018, 2019, and 2022 ABoVE airborne campaigns. We summarize the ~80 SAR flight lines and how they fit into the ABoVE experimental design. We provide hyperlinks to extensive maps, tables, and every flight plan as well as individual flight lines. We illustrate the interdisciplinary nature of airborne SAR data with examples of preliminary results from ABoVE studies including: boreal forest canopy structure from tomoSAR data over Delta Junction, AK and the BERMS site in northern Saskatchewan and active layer thickness and soil moisture data product validation. This paper is presented as a guide to enable interested readers to fully explore the ABoVE L- and P-band SAR data.

**Short Summary**

NASA's Arctic Boreal Vulnerability Experiment (ABoVE) conducted airborne synthetic aperture radar (SAR) surveys of over 120,000 km2 in Alaska and northwestern Canada during 2017, 2018, 2019, and 2022. This paper summarizes those results and provides links to details on ~80 individual flight lines. This paper is presented as a guide to enable interested readers to fully explore the ABoVE L- and P-band SAR data.

**Keywords**: Airborne Synthetic Aperture Radar (SAR), Interferometric SAR (InSAR), Polarimetric SAR (PolSAR), Tomographic SAR (tomoSAR), Arctic, tundra, taiga, boreal forest, permafrost, Arctic Boreal Vulnerability Experiment (ABoVE)

**1 Introduction**

The Arctic region contains a remarkable diversity of cold-adapted biota, habitats, and permafrost-affected ecosystems [McGuire 2009; Vincent 2011]. As with other components of the Arctic system, Arctic ecosystems are strongly interdependent and the rapid degradation of the Arctic cryosphere is altering their physical, biogeochemical, and biological linkages in ways that may be irreversible [Vincent 2011; Hinzman 2013]. Understanding characteristics that make Arctic ecosystems vulnerable or resilient to this change is the overarching objective of NASA's Arctic Boreal Vulnerability Experiment (ABoVE, https://above.nasa.gov/). Miller et al. [2019] describes how airborne campaigns fit into the broader ABoVE research strategy and how the foundational synthetic aperture radar (SAR) measurements formed the framework around which all other airborne data acquisitions were planned.

**Figure 1. Flight lines for the L-band and P-band PolInSAR measurements capture critical bioclimatic, permafrost, and geographic gradients as well as key field sites and long-term measurement records across the 4 Mkm$^2$ ABoVE domain. The flight lines are collected into 10 composites which roughly correspond to the Alaskan (A1-A4) and Canadian (C1-C6) regions sampled on individual flight days. © Google Maps**





ABoVE SAR flight lines (Figure 1) were planned to leverage legacy L- and P-band SAR transects
acquired during the pre-ABoVE period; remotely-sensed permafrost active layer thickness time series
derived from satellite interferometric SAR observations (ReSALT) [Schaefer 2015]; SAR data from
PALSAR, PALSAR-2, RadarSat, RadarSat-2, and Sentinel-1; historic or planned airborne LiDAR
acquisitions; and data from existing field sites [Hoy 2018]. Legacy airborne SAR flight lines include the
L-band grid acquired over the Boreal Ecosystem Research and Monitoring Sites (BERMS) area near
Prince Albert, SK during SMAP CanEx 2010 [Magagi 2012], the P-band lines over the BERMS area
acquired from 2012-2015 during the Airborne Microwave Observatory of Subcanopy and Subsurface
(AirMOSS) Earth Ventures Sub-orbital (EV-S1) investigation [Allen 2010; Moghaddam 2016], and a
collection of 10 L- and P-band flight lines acquired over the Seward Peninsula, Northwestern Interior,
and North Slope of Alaska during 2014 and 2015 [Chen 2019a, 2019b]. The BERMS area observations,
in particular, link ABoVE to the Boreal Ecosystem–Atmosphere Study (BOREAS) studies of the 1990s
[Sellers 1995; 1997].

Hoy et al. [2018] compiled information on more than 6,700 field sites and previous remote sensing data
sets to help plan the SAR flight lines and the ABoVE Airborne Campaigns [Miller 2019]. This
compilation is intended to help investigators understand flight line choices and identify ground locations
used to anchor individual flight lines. SAR data users may also search for the underlying data available
within each flight line. Key anchor points for the SAR flight lines include: Active layer thickness
measurements from the Circumpolar Active Layer Monitoring network (CALM); Permafrost
temperatures and annual thaw depths from the Global Terrestrial Network for Permafrost (GTN-P)
database; Soil moisture and permafrost state data from the Department of Energy's Next Generation
Ecological Experiment-Arctic (NGEE-Arctic) field sites on the Seward Peninsula and near Utqiaġvik
(formerly Barrow), AK; Extensive in situ terrestrial and aquatic ecosystem data as well as airborne
LiDAR and spectral imagery from NSF's National Ecological Observatory Network (NEON) D18
tundra field sites near Utqiagvik (Barrow), AK and Toolik Lake, AK, and from the D19 taiga field sites
near Caribou/Poker Creek, AK, Delta Junction, AK, and Healy, AK; Detailed ecological and physical
climate time series from NSF's Long Term Ecological Research (LTER) Arctic (Toolik Lake) and
Boreal Forest (Bonanza Creek) sites; Long-term boreal forest inventory data from the Canadian
Forestry Service's (CFS) Climate Impacts on Productivity and Health of Aspen (CIPHA) and High
Elevation & Latitude Climate Change Impacts & Adaptation (HELCIA) plots; and Long term
permafrost, hydrology and ecology time series records from the Canadian Changing Cold Regions
Network (CCRN) sites at Trail Valley Creek, NWT, Havikpak Creek, NWT, Scotty Creek, NWT,
Baker Creek, NWT, Wolf Creek Research Basin, YT, and the BERMS site at White Gull Creek, SK.

Airborne SAR data enable numerous ecosystem and ecosystem change research investigations [NRC
2014]. ABoVE researchers are using the airborne L- and P-band data to: Quantify permafrost active
layer thickness and soil moisture content [Bakian-Dogaheh 2020]; Complement AirSWOT Ka-band
acquisitions to determine water surface elevations in Arctic lakes, wetlands, and rivers [Pitcher
2019a,b]; Investigate boreal forest and tundra fire scars, especially in conjunction with fire disturbance
plots [Tank 2018; Walker 2018 a,b; 2019a,b; French 2020; Holloway 2020; Loboda 2021]; Map tree
density and distribution across the Tundra-Taiga ecotone; Provide control point data for the ArcticDEM



[Porter 2018; Meddens 2018]; Investigate lidar-radar fusion remote sensing for boreal forest
characterization as a precursor to NISAR/IceSAT-2 investigations [Silva 2021]; Quantify expansion and
sediment mass flow from massive retrogressive thaw slumps – so-called megaslumps – on the Peel
Plateau along the Dempster Hwy west of Fort McPherson [Kokelj 2013; 2015]; Classify Arctic
wetlands and habitats [French 2020]; and Support algorithm development for NISAR (L-band) and
BIOMASS (P-band) estimates of boreal forest structure and above ground biomass [Quegan 2019;
Saatchi 2019]. Goetz [2021] summarizes how the ABoVE airborne SAR data are helping advance
Arctic-boreal understanding and the remaining knowledge gaps still to be addressed.
This paper presents an annotated guide to enable interested readers to fully explore the ABoVE L- and
P-band SAR data acquired during the 2017, 2018, 2019, and 2022 ABoVE airborne campaigns. Section
2 provides details on the L- and P-band SAR instruments and the flight line catalog. Section 3
summarizes the daily sorties from each airborne campaign. Section 4 briefly describes the tomographic
SAR (tomoSAR) experiments flown over Delta Junction, AK and the BERMS site near Prince Albert,
SK. Section 5 describes some of the ABoVE SAR data products and their validation. Section 6
highlights the synergies between the L- and P-band airborne SAR data and other airborne sensors.
Section 7 summarizes access to the data products. Section 8 discusses potential future acquisitions and
outlooks for exploiting these data. Additionally, we include an Appendix which describes the ~80
ABoVE SAR flight lines and how each line fits into the ABoVE experimental design. The Appendix
also provides extensive maps and tables for every flight plan and individual flight lines as well as a list
of the acronyms and abbreviations. The Supplemental Information includes hyperlinked versions of the
tables for direct access to flight lines and flight plans.
**2 The L-Band and P-band Airborne SAR Instruments and Data Acquisition**
Both the L- and P-band airborne SARs are sensitive to geometrical and material properties of
vegetation, soil surface, and subsurface profiles [Saatchi and Moghaddam 2000; Tabatabaeenejad 2011;
Tabatabaeenejad 2015]. The joint use of both L- and P-band gives enhanced sensitivity to near-surface
(< 5 cm, L-band) and root zone (10-40 cm, P-band) portions of the subsurface profile compared to use
of either wavelength alone [Du 2015]. Airborne acquisitions with both SARs provide 6-10 m spatial
resolution, ~15 km swaths and transect lengths of 100 – 200 km, making them ideal for surveying
above-ground biomass and vegetation canopy structure [Hensley 2014; 2016] as well as the tundra-taiga
ecotone [Montesano 2016]. Special tomoSAR data were acquired over the well characterized BERMS
site in northern Saskatchewan and the NEON site in Delta Junction, AK to quantify the performance of
both SARs in reproducing the structure and biomass of boreal forests.

**2.1 The L-band SAR Instrument**
NASA's airborne L-band SAR (initially named the Uninhabited Aerial Vehicle Synthetic Aperture
Radar (UAVSAR) system) is a compact pod-mounted polarimetric instrument for interferometric
repeat-track observations that was developed at the NASA Jet Propulsion Laboratory (JPL) and Dryden



Flight Research Center (DFRC) in Edwards, CA. It operates at a center frequency of 1.2575 GHz (wavelength = 23.8 cm) with 80 MHz bandwidth. It is deployed on a Gulfstream III aircraft and images the Earth surface from a nominal 12.5 km altitude. The image swath is collected off-nadir in a ~22 km wide with incidence angles ranging from ~22°–67°. The instrument spatial resolution is 0.8m (along flight-line) by 1.7m (slant range, along line-of-sight (LOS) from the antenna to the ground). Topographic information is derived from phase measurements that, in turn, are obtained from two or more passes over a given target region. Its 1.26 GHz frequency results in radar images that are well-correlated from pass to pass. Polarization agility facilitates terrain and land-use classification.

All L-band SAR data are publicly available at http://uavsar.jpl.nasa.gov/ as individual InSAR products or as a single look complex (SLC) stack product of coregistered images for individual flight lines. Products are also available from the UAVSAR data portal at the Alaska SAR Facility Distributed Active Archive Center (https://asf.alaska.edu/data-sets/sar-data-sets/uavsar/). The L-band SAR provides key pre-launch algorithm development and validation data sets [Saatchi 2019] for the NASA-ISRO SAR (NISAR) mission [Rosen 2017].

**2.2 The P-band SAR Instrument**

The P-band SAR was developed circa 2012 for the Earth Ventures Sub-orbital (EV-S1) Airborne Microwave Observatory of Subcanopy and Subsurface (AirMOSS) investigation [Allen 2010; Moghaddam 2016]. The radar is based on JPL's L-band UAVSAR system. The P-band SAR inherits UAVSAR's existing L-band RF and digital electronics subsystems. New up- and down-converters convert the L-band signals to UHF frequencies (280-440 MHz). The passive antenna is based on the legacy GeoSAR design [Chapin 2012].

The P-band SAR was flown more than 1200 h from 2012 to 2015, covering regions of 2500 km$^2$ spread over nine major biomes in North America during the AirMOSS EV-S1 investigation [Tabatabaeenejad 2020]. Legacy acquisitions in Alaska [Chen 2019a, b] and over the BERMS site in northern Saskatchewan [Chapin 2012, 2018] provide an opportunity for extended time series analysis. All P-band SAR data are publicly available at http://uavsar.jpl.nasa.gov/. Additionally, ABoVE P-band SAR data will provide valuable insights into the characterization of boreal forest and tundra ecosystems by the upcoming BIOMASS mission [Le Toan 2011; Quegan 2019].

**2.3 The Platform Precision Autopilot (PPA) System**

To support cm-precision interferometric land surface characterization, repeat pass measurements acquired by the SARs need to be taken from flight paths that are nearly identical. Both the L- and P-band SARs utilize real-time GPS that interfaces with the platform flight management system (FMS) to confine the repeat flight path to within a 10 m tube over a 200 km course in conditions of calm to light turbulence. The FMS is also referred to as the Platform Precision Autopilot (PPA). Additionally, the radar vector from the aircraft to the ground target area must be similar from pass to pass. This is



accomplished with an actively scanned antenna designed to support electronic steering of the antenna
beam with a minimum of 1º increments over a range to exceed ±15º in the flight direction.
ABoVE SAR measurements were typically acquired with platform RMS deviations less than ±3 m. Any
platform deviations larger than ±10 m from the programmed flight path resulted in the acquisition being
terminated and a real time decision made to reacquire the line from the beginning or to continue with
the flight plan and proceed to the next line. This decision balanced the science priority of the flight line,
fuel consumption and remaining endurance, the number of flight lines yet to be acquired in the day's
flight plan, distance from our base of operations, and whether there would be future opportunities to
collect a given line by adding it to an upcoming flight plan. The flight team was extremely efficient in
executing these decisions, resulting in 95% flight line acquisition success across the 2017-2019 period.
**2.4 Airborne SAR Flight Line and Flight Plan Designations**
The JPL SAR team devised a convenient and powerful way to identify airborne SAR data acquisitions
for the Facility and PI instruments under their charge. Each L-band or P-band SAR flight line receives a
unique 5-digit identifier consisting of the three-digit GPS compass heading followed by a two-digit
index.  A 6-character text string is also associated with each line for ease of identification. The text
string proceeds the numerical ID and usually provides abbreviated geographic or infrastructure
information that characterizes the line. For example, L-band flight line Teller_04901 identifies the flight
line on the Seward Peninsula that overflies the NGEE-Arctic Teller watershed. The flight line identifier
is a constant and, once assigned, is used whenever a line is reflown. In some cases, there are
overlapping or nearly identical flight lines which differ slightly in their ID number. L-band and P-band
flight lines use the same flight line identification system, allowing rapid identification of overlapping L-
and P-band data acquisitions.
Flight Plans are assembled from the composite flight lines for a given sortie. Each flight plan also
receives a unique 5-digit identifier based on the year flown (digits 1 and 2) and the flight number for
that year (digits 3-5). For example, L-band flight plan 17093 was flown in 2017 and was the 93rd sortie
flown that year. Note that there may be more than one sortie flown on a given day, in which case each
would have a unique flight plan identifier even though they were flown on the same calendar day and
may include some or all of the same flight lines.
In the Supplemental Information we provide hyperlinks to the JPL UAVSAR data portal
(https://uavsar.jpl.nasa.gov/cgi-bin/data.pl). This provides links to the individual flight line data, maps,
and related flight plans that acquired data over one or more of the individual flight lines. We hope this
enables interested readers to explore the ABoVE L- and P-band SAR data more fully. These data and all
other airborne data from the ABoVE campaigns may be explored on NASA's EarthData ABoVE Portal
(https://search.earthdata.nasa.gov/portal/above/search). Ground sites used to design the orientation and
locations of the flight lines are archived at the ORNL DAAC [Hoy 2018].


Earth System Discussions
Science
Data

## 3 The ABoVE Airborne SAR Campaigns

The L-band (Figure 2) and P-band (Figure 3) SARs were considered foundational measurements in the ABoVE airborne campaign strategy [Miller 2019]. The ~80 flight lines described in the Appendix formed the framework for the remainder of the airborne remote sensing acquisitions. The ABoVE SAR strategy was to execute same day acquisitions of both L- and P-band flight lines (Figure 1) for a given sortie during 2017 to optimize dual frequency retrievals; however, technical issues forced us to fly the instruments sequentially. The baseline L-band campaigns were flown in June (DOY 164-173) and September (DOY 251-263) of 2017 to characterize the land surface during periods of minimum and maximum active layer thickness, respectively. Subsequent L-band campaigns in 2018 (DOY 231-241), 2019 (DOY 247-260) and 2022 (DOY 226-237) provide a time series synched to maximum annual active layer thickness. P-band campaigns were conducted in May-June (DOY 142-157) and August (DOY 219-227) of 2017. There was a 2-day P-band mini-campaign in October 2017 to extend the legacy time series of early cold season acquisitions over the Seward Peninsula, NW Alaska and North Slope Alaska (DOY 280-283).

**Figure 2. Sahtu students Mandy Bayha (front left) and Joanne Speakman (front center) pose with their mentor Cindy Gilday (front right) and NASA flight crew in Yellowknife, NT after completing a L-band SAR survey flight around the Great Slave Lake**



70  Region on 22 August 2018 (Flight Plan 18048). This experience gave these Northerners a new appreciation for how NASA was
71  helping understand, preserve, and protect their lands. Photo Credit: Stephen M. Fochuk, Government of Northwest Territories.



Figure 3. The P-band SAR team with the NASA JSC G-III (N992NA) on the tarmac in Fairbanks, AK on 18 August 2017 after
completing a survey of the Upper Mackenzie Valley (Flight plan 17083). Photo Credit: M. Moghaddam.

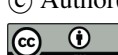



### 3.1 Alaskan Flight Lines

The Alaskan SAR flight lines are broken into four main regional collections: A1) North Slope Alaska,
A2) Seward Peninsula and Northwest Alaska, A3) Eastern Interior, and A4) Southwest Alaska and the
Yukon-Kuskokwim Delta (**Figure 1**). Individual flight
lines were planned based on long-term ground
monitoring sites [Hoy 2018], existing or planned field
research, recent disturbances, important geographic or
ecological gradients, complementary remote sensing
data, and consultation with indigenous peoples and
governments [Miller 2019]. Legacy L- and P-band flight
lines from the AirMOSS EV-S1 investigation [Allen
2010; Moghaddam 2016] in the Seward Peninsula, NW
Alaska, and the North Slope were adapted for ABoVE
use. Acquisition of P-band flight lines in the central
Interior was not possible due to a military radar keep-out
zone centered near Clear, AK. The keep-out zone is
shown in all P-band flight plan maps (Ex. **Figure 4**).

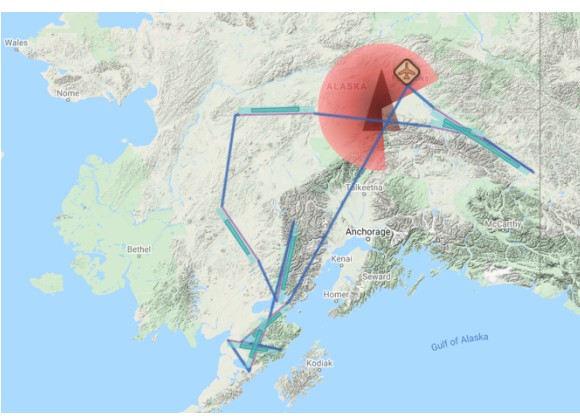

**Figure 4.** The military radar at Clear, AK creates a large P-band operations keep-out zone in the central Interior (red areas). The aircraft symbol marks our Fairbanks International Airport (PAFA) base of operations. Data acquisitions (blue bars) are from Flight Plan 17054. © Google Maps

### 3.2 Canadian Flight Lines

The Canadian SAR flight lines are broken into six regional collections: C1) Lower Mackenzie Valley
and Northern Yukon Territory, C2) Southern Yukon Territory, C3) Upper Mackenzie Valley, C4) Great
Slave Lake Region, C5) Transboundary Watershed, and C6) Southern Boreal Forest/BERMS.
Individual lines were planned based on long-term ground monitoring sites [Hoy 2018], existing or
planned field research, recent disturbances, important geographic or ecological gradients,
complementary remote sensing data, and consultation with local inhabitants and governments [Miller
2019]. Legacy L- and P-band flight lines in the BERMS area from the CANEX 2010 campaign [Magagi
2012] and the AirMOSS EV-S1 investigation [Chapin 2012, 2018] provide the potential to establish
longer time series.
Flight planning for the Canadian transects benefited tremendously from consultations with our
Canadian colleagues and interested parties in Yellowknife, NT and Whitehorse, YT in 2015 and 2016.
Extensive discussions with the Government of the Northwest Territories (GNWT), the Government of
the Yukon Territory, First Nations representatives, and scientists from Polar Knowledge Canada
(POLAR), the NWT Center for Geomatics, and the Canadian Forestry Service (CFS) Northern Forestry
Centre (NoFC) were critical to designing a strategy that captured many of their observing priorities.
Subsequent discussions in Yellowknife during 2017 and 2018 enabled us to disseminate preliminary
results and coordinate the flights with same-day field data acquisitions.




## 4 TomoSAR Measurements of Boreal Forest Structure

SAR tomographic methods have proven extremely adept at measuring vegetation vertical structure at a
variety of wavelengths including L- and P-bands. The three-dimensional vegetation structure and its
changes resulting from either natural or anthropogenic causes are key ecosystem monitoring parameters.
ABoVE collected tomographic L- and P-band SAR data over the boreal forest near Delta Junction, AK
in September of 2017. UAVSAR (L-band) and the German Space Agency's F-SAR (L- and S-bands)
acquired coordinated tomographic SAR data at the BERMS site near Saskatoon, SK in August 2018.
Ground truth data sets and LiDAR data from the NASA LVIS system were also acquired at BERMS in
2017 [Blair 2018]. We compared L- and P-band tomography at Delta Junction and L-band and S-band
tomography from the two systems, to each other, and to the LiDAR data sets at BERMS. Here we
provide a preliminary analysis of the data acquired at BERMS.

BERMS is a southern boreal forest site with gentle topography dominated by Jack Pine and Aspen
stands. There is active logging in the area and the site contains clear cut areas and new growth stands in
various maturity states. The tomography data acquisition at BERMS was planned jointly in cooperation
with the German Space Agency (DLR) who flew the F-SAR radar and acquired data at L-band and S-
band. The UAVSAR and F-SAR flight lines were designed to overlap each other and LVIS data
acquired at the site in 2017. LVIS reacquired BERMS area data again in 2019 with the LVIS-F and
LVIS-C instruments [https://lvis.gsfc.nasa.gov/Data/Maps/ABoVE2019Map.html]. Figure 5 (left)
shows swaths for the UAVSAR and F-SAR radars along with the LVIS data. UAVSAR acquired L-
band tomography data on a racetrack pattern to get multiple incidence angle data for most points in the
swath. Because UAVSAR and F-SAR fly at 12500 mAGL and 4200 mAGL, respectively, it is not
possible to acquire data with the same incidence angles across the swath. Thus, we configured the flight
lines to overlap so that the 40° incidence angle points would coincide. Figure 5 (right) shows photos
collected at four of our seventeen ground truth sites during the tomoSAR acqusitions.

LVIS full waveform LiDAR provides surface elevations and tree height estimates as well as LiDAR
echo strength throughout the canopy and thereby information on the canopy internal structure. From
LVIS waveforms many products are possible including surface elevation, tree height, moments of the
returned waveform distribution and cumulative percentile elevations. We compared these waveforms to
radar tomographic profiles for the different radar wavelengths.

BERMS field measurements consist of soil moisture measurements at the 17 sites using the average of
15 measurements distributed over 60 m × 60 m plots on the day of the UAVSAR radar observations. At
BERMS the soil was very dry, roughly 10% volumetric soil moisture or less, during the radar
observations. During the summer of 2020 diameter at breast height (DBH) measurements, used to
estimate biomass, for a subset of our selected sites was planned but postponed due to COVID-19.
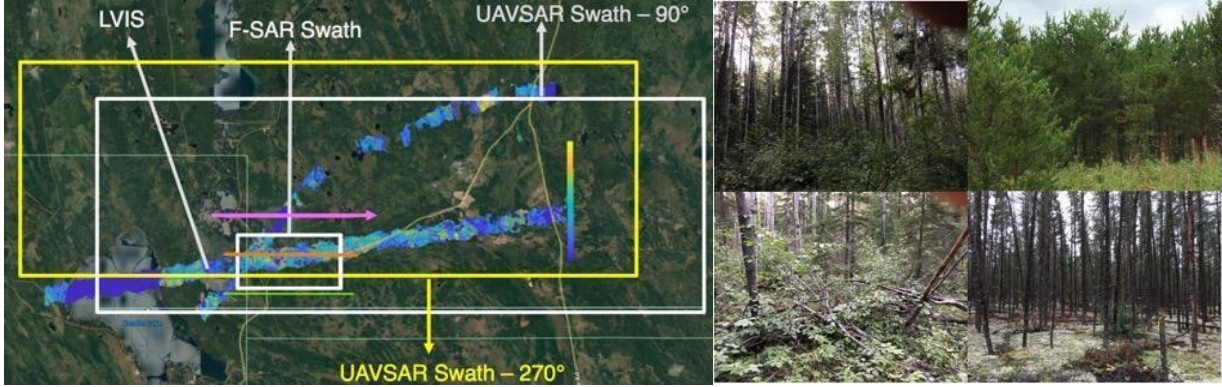

**Figure 5.** LEFT: Experimental design for the BERMS area TomoSAR flights in August 2018. The large White and yellow boxes show the ~18 km-wide UAVSAR L-band swaths. The offset is due to the off-nadir viewing angle of the L-band SAR – it is pointed to the south when flying the 270º swath and pointed north when flying the 90º swath. The small white box near the center of the image marks the ~3 km-wide F-SAR swath. LVIS LiDAR data are the ~1.5 km-wide colored swaths across the image; the color scaling reflects the canopy height. RIGHT: Photos from four of the 17 plots used for in situ ground truth measurement at BERMS. Vegetation at BERMS was mostly Jack Pine and Aspen with many areas having dense understory vegetation. Most areas have substantial detritus and ground litter left over from previous logging operations. © Google Earth

**Figure 6** compares UAVSAR L-band tomography with F-SAR L-band and S-band tomography and LVIS LiDAR data along a transect shown as a yellow line on the right in the figure. Tree height along the transect varied from 10-20 m. Middle of Figure 2 is the UAVSAR L-band transect. Top of the figure shows the LVIS RH25m RH50, RH75 and RH95 profiles overlaid on the UAVSAR tomogram and below are the radar and LiDAR vertical profiles. On the left of Figure 2 are the F-SAR L-band and S-band tomographic profiles along with the LVIS RH100 data. The L-band radar profiles exhibit power concentrated at the base of the canopy whereas the LVIS LiDAR data show more return from the middle portion of the canopy. S-band obtains greater returns in the upper canopy compared to L-band and show more uniform scattering within the canopy.

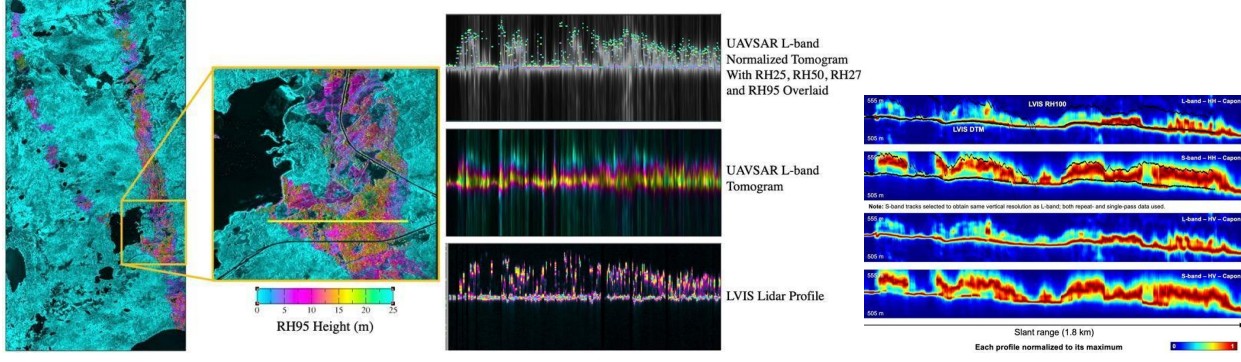

**Figure 6.** On the left shows location of transect as yellow line overlaid on UAVSAR imagery in grayscale and LVIS RH95 data is color where UAVSAR L-band, DLR F-SAR L and S-band and LVIS LiDAR profiles are compared. Center figure shows UAVSAR L-band tomographic profiles along the transect along with the corresponding LVIS LiDAR profiles. On the right are the corresponding F-SAR L and S-band tomographic profiles. The L-band radar profiles exhibit power concentrated at the base of the canopy whereas the LVIS LiDAR data show more return from the middle portion of the canopy. S-band obtains greater returns in the upper canopy compared to L-band and show more uniform scattering within the canopy.



## 5 ABoVE SAR Data Products

Here we highlight some data sets enabling or derived from the ABoVE L- and P-band airborne SAR acquisitions. They represent the current state of the art for the study of permafrost-affected ecosystems using SAR. The ABoVE science team continues to develop additional products and the insights from these studies will be published separately. Links to the repositories for each of these data sets are provided below.

### 5.1 Active Layer Thickness (ALT)

The Permafrost Dynamics Observatory (PDO) data product estimates seasonal subsidence, active layer thickness (ALT), soil Volumetric Water Content (VWC), and uncertainties at 30-m resolution for 66 flight lines across Alaska and Northwest Canada [Michaelides 2021; Chen 2021a,b]. The PDO retrieval uses L-band Synthetic Aperture Radar (SAR) data acquired by the Uninhabited Aerial Vehicle Synthetic Aperture Radar (UAVSAR) instrument and P-band data acquired by the Airborne Microwave Observatory of Subcanopy and Subsurface (AirMOSS) instrument. The PDO results for each flight line appear in separate netcdf files. Each line has a spatial resolution of 30 meters on the ABoVE common grid with a width of 22 km based on the swath width of the AirMOSS instrument. The flight lines as a whole cover many ecosystem types and provide north-south and east-west gradients in ALT and soil moisture across the ABoVE domain (https://daac.ornl.gov/cgi-bin/dsviewer.pl?ds_id=1796).

**Table 1** defines all variables in PDO data files. The first eight variables represent the four primary outputs of the PDO algorithm and associated uncertainties. Subsidence and ALT represent one-time measurements for the 2017 thaw season [Schaefer 2021]. VWC, defined as the ratio of water volume to total soil volume, represents soil moisture at maximum thaw for 2017. We assume a vertical profile of VWC and estimate Sw0 and wtd, the parameters that define the exact shape of the assumed profile. The product includes a Python script that will create a map of VWC averaged over any user specified depth range. We included maps of VWC averaged over depth ranges of hand-held soil moisture probes commonly used in ABoVE fieldwork.

Table 1. Variables in the Permafrost Dynamics Observatory (PDO) data files

| Variable | Full Name | Units | Description |
|---|---|---|---|
| alt | Active Layer Thickness | M | Maximum thaw depth at the end of summer |
| sub | Subsidence | M | Surface subsidence from start of thaw after snow melt to maximum thaw depth in August or September |
| Sw0 | Surface Saturation Fraction | m3/m3 | The ratio of water volume to pore space volume at the surface or zero meters depth |
| wtd | Water Table Depth | M | The depth from the surface to the level where the soil is 100% saturated |
| alt_unc | Uncertainty ALT | M | Uncertainty of estimated ALT |



| sub_unc | Uncertainty Subsidence | M | Uncertainty of estimated seasonal subsidence |
|---|---|---|---|
| Sw0_unc | Uncertainty Surface Saturation Fraction | m3/m3 | Uncertainty of estimated surface water saturation fraction |
| wtd_unc | Uncertainty Water Table Depth | M | Uncertainty of estimated water table depth |
| mv_6cm | VWC from 0 to 6 cm | m3/m3 | The ratio of water volume to soil volume averaged over zero to 6 cm depths |
| mv_12cm | VWC from 0 to 12 cm | m3/m3 | The ratio of water volume to soil volume averaged over zero to 12 cm depths |
| mv_20cm | VWC from 0 to 20 cm | m3/m3 | The ratio of water volume to soil volume averaged over zero to 20 cm depths |
| mv_alt | VWC from 0 to ALT | m3/m3 | The ratio of water volume to soil volume averaged over the entire active layer, from zero to ALT |

**5.2 Alaska Active Layer and Soil Moisture Properties from Airborne P-band SAR**
Chen et al. [2019b] synthesized the P-band polarimetric synthetic aperture radar (PolSAR) data
collected in August and October of 2014 and 2015 during the AirMOSS EV-S1 investigation with the
ABoVE P-band measurements collected in August and October of 2017 to estimate soil geophysical
properties over 12 study sites in Northern Alaska (see Figure S2). Soil properties reported include the
ALT, soil dielectric constant, soil moisture profile, surface roughness, and their respective uncertainty
estimates at 30-m spatial resolution (https://doi.org/10.3334/ORNLDAAC/1657).
Most of the study sites are located within the continuous permafrost zone and where the aboveground
vegetation consisting mainly of dwarf shrub and tussock/sedge/moss tundra has a minimal impact on P-
band radar backscatter. These data were used as inputs to the L-band ReSALT data described in Section
19 5.1.

**5.3 In Situ Soil Moisture and Thaw Depth Measurements**
In situ measurements of soil moisture, thaw depth, and other quantities are essential to calibrate and
validate ABoVE SAR retrievals. The ABoVE team established a set of standardized measurement
protocols for field plots to ensure uniform data products and quality in measurements collected by
different groups across the ABoVE domain and across multiple years. Numerous teams collected in situ
data during the initial 2017 Airborne Campaign, with more targeted field acquisitions conducted in
2018 and 2019 [Bourgeau-Chavez et al. 2019a,b, 2021; Bakian-Dogaheh 2020; Loboda 2021].
Bourgeau-Chavez and coworkers [2019a,b; 2021] collected soil moisture at 6, 12, 20, and 50 cm depths,
ALT, soil profiles and biophysical measurements of aboveground canopy and ground layers in the
Greater Slave Lake Region (C4). These data provide vegetation community characteristics and
biophysical data collected in 2018 from areas that were burned by wildfire in 2014 and 2015, and from
nine unburned validation sites. Vegetation data include vegetation inventories, ground cover, regrowth,





tree diameter and height, and woody seedling/sprouting data at burned sites, and similar vegetation
community characterization at unburned validation sites. Additional measurements included soil
moisture, collected for validation of the UAVSAR airborne collection, and depth to frozen ground at the
nine unburned sites. This 2018 fieldwork completes four years of field sampling at the wildfire areas.
Bakian-Dogaheh et al. [2020] measurements included soil dielectric properties, temperature, and
moisture profiles, active layer thickness (ALT), and measurements of soil organic matter, bulk density,
porosity, texture, and coarse root biomass from the surface to permafrost table in soil pits at selected
sites along the Dalton Highway in Northern Alaska (A1). Their investigation sites included Franklin
Bluffs, Sagwon, Happy Valley, Ice Cut, and Imnavait Creek. Measurements collected at Franklin Bluffs
were concurrent with an August 2018 ABoVE L-band flight.
(https://doi.org/10.3334/ORNLDAAC/1759).
Loboda et al [2022] collected field measurements from unburned sites and single and repeated burns in
the Noatak River valley and the Seward Peninsula regions of the Alaska tundra in July-August in the
years 2016-2018. The data include ocular assessment of vegetation cover, soil moisture at 6 and 12 cm,
soil temperature at 10 cm, organic soil thickness, thaw depth, and weather measurements.
(https://doi.org/10.3334/ORNLDAAC/1919)
The strong partnership between the ABoVE and NGEE-Arctic projects also resulted in coordinated
same-day acquisition of airborne L- and P-band SAR data with in situ soil moisture and thaw depth
measurements over the NGEE-Arctic study site at Barrow (Utqiagvik), AK and the Seward Peninsula
watersheds near Teller, AK, Council, AK, and Kougarok, AK [Wilson 2018]. These data provide
critical calibration for the ABoVE SAR retrievals under continuous (Utqiagvik) and discontinuous
(Seward Peninsula) permafrost conditions. Version 2 (V2) of the in situ soil moisture and thaw depth
measurements covering years 2017-2019 was released in November 2020.
(https://doi.org/10.5440/1423892)
**6 Synergy with Other Airborne Sensors**
Miller et al. [2019] described the overall ABoVE Airborne Campaign design strategy and anticipated
airborne sensor synergies. Here, we highlight three SARs and a LiDAR – AirSWOT (NASA), F-SAR
(DLR), LS-ASAR (ISRO) and LVIS (NASA) – whose acquisitions in the ABoVE domain were
specifically designed to complement and leverage the ABoVE L- and/or P-band SAR acquisitions.
Many other airborne sensor synergies are being exploited by the ABoVE science team and are reported
separately.
**6.1 AirSWOT**
NASA's AirSWOT airborne instrument suite has been developed to support the Surface Water and
Ocean Topography (SWOT) mission. The heart of AirSWOT is the Ka-band SWOT Phenomenology
Airborne Radar (KaSPAR). KaSPAR collects two swaths of across-track interferometry data: one swath





from nadir to 1 km and a second swath that extends from 1 km to 5 km off-nadir. AirSWOT flight lines
for ABoVE were designed to center the AirSWOT swath on the center of the P-band swath for
maximum overlap. KaSPAR is complemented by a high-resolution color-infrared (CIR) Digital Camera
System [Kyzivat 2019a,b] and a Precision Inertial Measurement Unit (IMU) for accurate attitude and
positioning information. In 2015 AirSWOT made pre-ABoVE deployments to the Tanana River Valley
[Altenau 2017] and the Yukon Flats [Pitcher 2019a, 2019b] in Region A3.

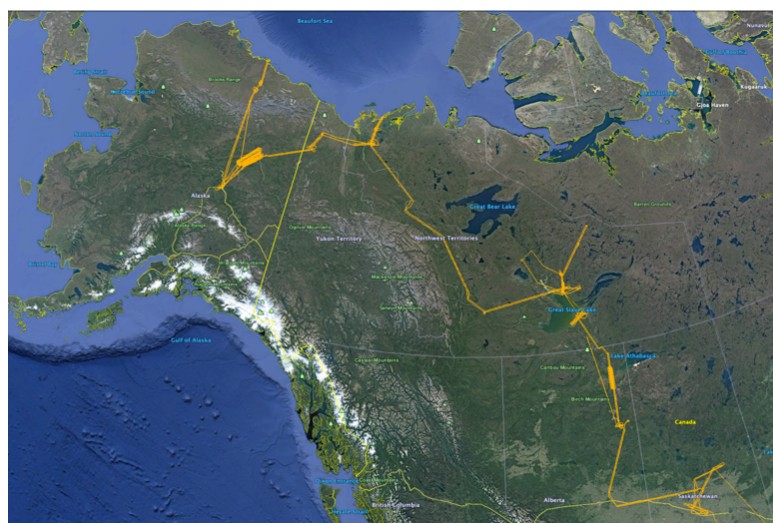

In 2017, AirSWOT deployed to acquire
early season (May-June) and late season
(August) WSEs across the ABoVE
domain. **Figure 7** shows the concentration
of AirSWOT lines in wetlands complexes
in the boreal forest, across the Canadian
Shield, along the Mackenzie River Valley,
and into the Arctic tundra. All of these
regions contain overlapping L- and P-
band acquisitions. Of special interest are
the lines in the Peace-Athabasca Delta
(36000: PADelE and 18035: PADelW)
and the Yukon Flats (21508: YFlatW,
21609: YflatE, and 04707: FtYuko) and
Trail Valley Creek, NT (01703: TukHwy)
where extensive on-water measurements
were made [Pitcher 2020]. Future joint
analyses of the Ka- and L-band data will
highlight the advances possible in pan-
Arctic hydrology from the upcoming NISAR and SWOT missions.

**Figure 7.** AirSWOT flight lines acquired during the 2017 ABoVE airborne campaign sampled wetlands ranging from the Arctic Ocean coast to the southern boreal forest. AirSWOT's Ka-band acquisitions were designed to overlap with the L- and P-band SAR near-field acquisitions (See Fig. 1). © Google Earth




## 6.2 F-SAR

The German Space Agency (DLR) developed the F-SAR instrument as an advanced airborne SAR testbed for technology and remote sensing applications [Reigber 2013]. F-SAR operates fully polarimetric at X-, C-, S-, L- and P-bands and features single-pass polarimetric interferometric SAR (PolInSAR) capabilities in X- and S-bands [Reigber 2013]. The radar covers an off-nadir angle range of 25 to 60 degrees and provides sub-meter scale spatial resolution from flight altitudes up to 6000 mAGL.

During August 2018 and April 2019, F-SAR was deployed to northern Canada as part of DLR's permafrost airborne SAR experiment (PermASAR). It was configured in X-, C-, S- and L–band mode and flew onboard a Dornier Do 228-212 research aircraft. Measurements were acquired from ~4500 mAGL. Coordinated tomoSAR transects were flown over the BERMS site in the southern boreal forest on 18 August (UAVSAR) and 23 August (F-SAR). Preliminary results [Hensley 2020] are summarized in Section 6. F-SAR also acquired data over the Scotty Creek watershed, flux towers, and AOIs (Figure 8), the Smith Creek flux tower (Wrigley, NT), Baker Lake, Havipak Creek, Trail Valley Creek, and Herschel Island, providing extensive opportunities to cross-compare F-SAR and the ABoVE SAR acquisitions.

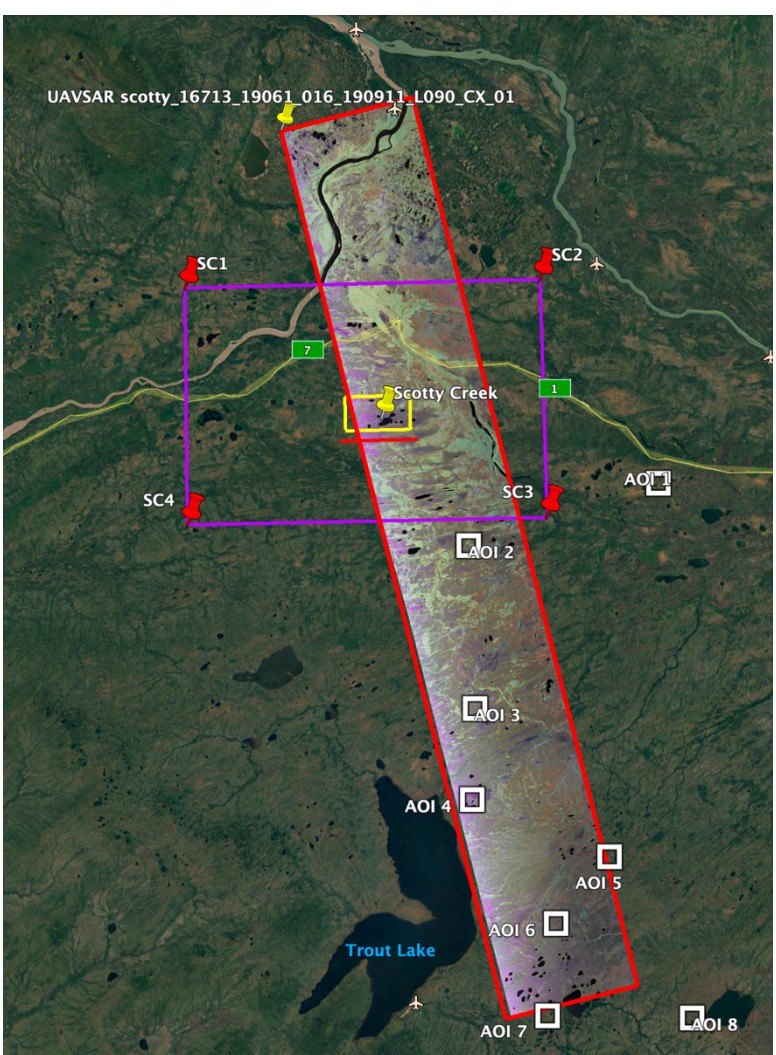

**Figure 8.** Overlap of the F-SAR acquisition at Scotty Creek, NT (yellow box) and the ABoVE L-band SAR line 16713 (red box & polarized SAR false color). The ABoVE line also captures Scotty Creek AOIs 2 – 6 (see Region C3 details, Sec. 5.3). The SAR data will complement and benefit from the extensive ground-based data acquired in this area [Quinton 2019]. © Google Earth





## 6.3 LS-ASAR (ISRO)

The Indian Space Research Organisation (ISRO) and NASA are jointly developing the NASA-ISRO Synthetic Aperture Radar (NISAR), which will map Earth's surface in L-band and S-band every 12 days [Rosen 2017]. As a precursor to the NISAR mission, ISRO has developed a L- and S-Band-Airborne SAR (LS-ASAR) to prepare the community to maximize the scientific and societal benefits of NISAR data [Ramanujam 2016; 2019; Mehra 2019]. LS-ASAR operates in Dual, Quad, and Hybrid and Polarization modes in both L- and S-bands. It covers incidence angles from 24°-77° with swaths ranging from 5.5 km to 15 km.

In December 2019 LS-ASAR flew a series of Arctic sea ice sorties from Fairbanks, AK. During this deployment, LS-ASAR also acquired data over a number of the ABoVE flight lines in Regions A1 (North Slope) and A3 (Eastern Interior) as well as over a number of glacier sites in the Alaska Range. The acquisitions are available via the NASA & ISRO ASAR Campaign page (https://uavsar.jpl.nasa.gov/cgi-bin/deployment.pl?id=L20191101) and are summarized in Figure 9. These data provide snow-on coverage that was a known deficiency of previous ABoVE airborne campaigns. Additionally, the LS-ASAR data extend coverage of these regions to S-band.

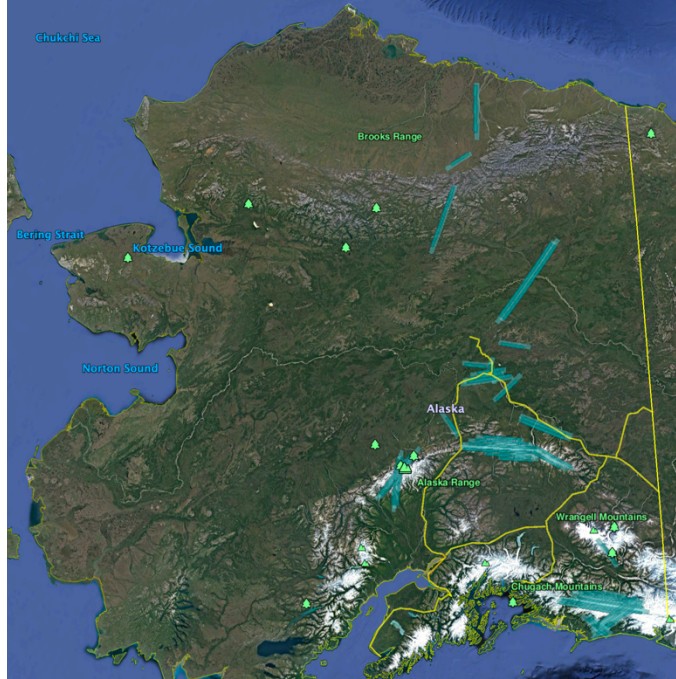

**Figure 9. L- and S-band SAR lines acquired over Alaska during the December 2019 ASAR Campaign. ASAR flight lines on the North Slope, in the Yukon Flats, and in the Western Interior exactly overlap ABoVE flight lines. These early winter acquisitions provide a preliminary look at cold season SAR data that will be explored in greater detail in the planned ABoVE-SnowEx campaign. © Google Earth**

## 6.4 LVIS

The Land, Vegetation, and Ice Sensor (LVIS) is an airborne, full waveform scanning laser altimeter which produces topographic maps with decimeter accuracy as well as vegetation vertical height and structure measurements [Blair 1999a,b]. Flight lines for LVIS (~1.4 km swath) were slaved to the centerline of the P-band swath during ABoVE, except where deviations were required to capture critical ground sites. LVIS-C (classic configuration) was deployed in 2017 aboard a B-200 and achieved limited coverage (Figure 10, left panel). During 2019, the new LVIS Facility instrument (LVIS-F) as well as LVIS-C were deployed on the NASA Gulfstream-V and achieved coverage of all legacy SAR lines (Figure 10, right panel).

LVIS' unique capability for measuring the sub-meter topography beneath boreal forest canopies
complemented the tomoSAR acquisitions over Delta Junction, AK and the BERMS site in northern
Saskatchewan [Hensley 2020; Section 6]. LVIS altimetry will also prove valuable in analyses of such
variables as permafrost degradation, active layer thickness, and water surface elevation; however,
LVIS' significantly narrower swath limits the spatial extent over which these analyses may be
performed.
In June-July 2017, the NASA LVIS Facility was deployed to sites in northern Canada and Alaska as
part of NASA's Arctic-Boreal Vulnerability Experiment (ABoVE) 2017 airborne campaign. During the
4-week deployment of LVIS-F, a total of 15 flights were flown over diverse science targets based out of
multiple airports in Canada and Alaska. Data are available in both Level1B and Level2 formats (**Table
2**). The Level1b data files contain the geolocated laser waveform data for each laser footprint. The
Level2 data files contain canopy top and ground elevations and relative heights derived from the
Level1b data. ABoVE LVIS L1B Geolocated Return Energy Waveforms, Version 1 [Blair and Hofton,
2018a] and L2 Geolocated Surface Elevation Product, Version 1 [Blair and Hofton, 2018b] may be
obtained from the National Snow and Ice Data Center via https://doi.org/10.5067/UMRAWS57QAFU
and https://doi.org/10.5067/IA5WAX7K3YGY, respectively.

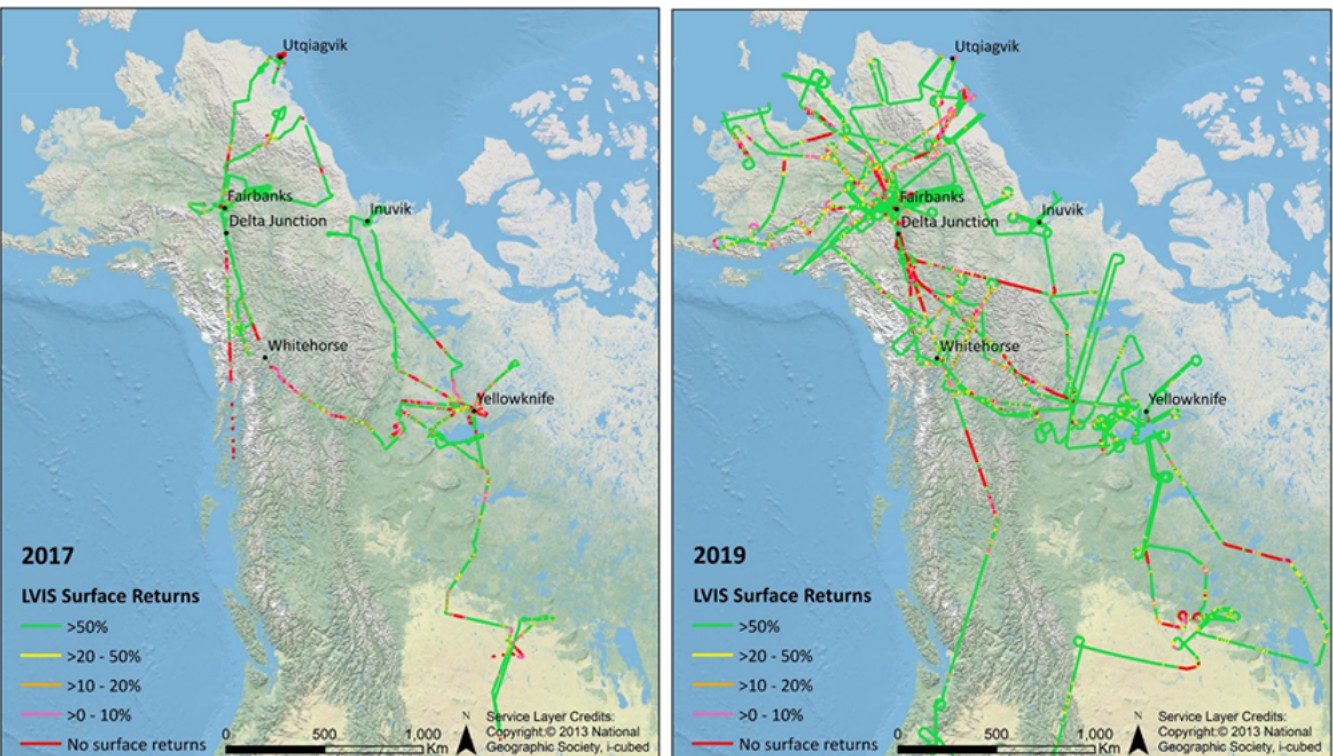

**Figure 10. Flight lines for the LVIS 2017 flights (left) and 2019 flights (right) were designed ot overlap with the near-filed portions
of the L-band and P-band SAR swaths to maximize opportunities for synergistic science. Aircraft and weather limited coverage
during the 2017 campaign, but complete coverage of the SAR flight lines was achieved in 2019. These data will enable studies of
SAR/LiDAR fusion over the Arctic-boreal regions as a precursor to NISAR/IceSat-2 studies. © National Geographic Society**




**Table 2. 2017 LVIS Data Products**

| LVIS Data Products | Format |
|---|---|
| Flight Trajectories | KMZ |
| Camera Trajectories | KMZ |
| LVIS L1A Camera Imagery | JPG* |
| LVIS L1B Geolocated Waveforms | HDF, LDS 2.0.2 |
| LVIS L2 Elevation and Height Products | ASCII TXT, LDS 2.0.2a |

In July-August 2019, the NASA LVIS Facility and LVIS Classic were deployed to sites in northern
Canada and Alaska as part of NASA's ABoVE 2019 airborne campaign. The increased range and
endurance of the Gulfstream-V platform enabled extensive sampling, including: all L-band SAR lines,
multiple IceSAT-2 underflights, and numerous ABoVE field sites. The available data products are given
in **Table 3**.
**Table 3. 2019 LVIS Data Products**

| LVIS Data Products | Format |
|---|---|
| Flight Trajectories | KMZ |
| Coverage Maps | KMZ |
| LVIS Classic L1B Geolocated Waveforms | HDF, LDS 2.0.3 |
| LVIS Classic L2 Elevation and Height Products | ASCII TXT, LDS 2.0.3 |
| LVIS Facility L1B Geolocated Waveforms | HDF, LDS 2.0.3 |
| LVIS Facility L2 Elevation and Height Products | ASCII TXT, LDS 2.0.3 |

**6.5 G-LiHT**
Zhao et al. [2022] used ABoVE airborne L- and P-band SAR to map boreal forest species and canopy
height in the Tanana Valley State Forest (TVSF) near Delta Junction, AK. They employed Random
Forests to train separate regression models for canopy height mapping and a classification model for
forest species mapping. Data derived from NASA's Goddard LiDAR, Hyperspectral, and Thermal
Imager (G-LiHT) system [Cook 2013] were treated as ground truth for the canopy height model
(CHM). Forest species prediction were referenced against (TVSF) Timber Inventory and Forest
Inventory and Analysis (FIA) data. The experimental results show the proposed method yields a root-
mean-square error of 1.90 m for forest height estimation and overall accuracy of 79.5% for forest
species classification. A significant finding was that PolSAR decomposition parameters, such as volume
scattering and entropy, strongly influenced the canopy height estimates. Interestingly, topography
played a crucial role in the species classification.



## 7 Data Availability

Links to the ABoVE L- and P-band SAR products, supporting data, derived products, and ancillary measurements are provided in the Appendix. Formal citations to all DOIs are provided in the References. The L- and P-band SAR data may be found at the JPL UAVSAR data portal, https://uavsar.jpl.nasa.gov/cgi-bin/data.pl. L-band data may also be accessed via the UAVSAR portal at the Alaska SAR Facility (ASF) DAAC (https://asf.alaska.edu/data-sets/sar-data-sets/uavsar/) while AirMOSS P-band data may be accessed via the ORNL DAAC, https://daac.ornl.gov/get_data/#projects, select "AirMOSS".

Miller et al. [2023; https://doi.org/10.3334/ORNLDAAC/2150] provides a detailed description of all 80 SAR flight lines and how each fits into the ABoVE experimental design. Extensive maps, tables, and hyperlinks give direct access to every flight plan as well as individual flight lines. It is a guide to enable interested readers to fully explore the ABoVE L- and P-band SAR data.

## 8 Summary

The ABoVE project conducted airborne L-band PolInSAR surveys in 2017, 2018, 2019 and 2022 across Alaska and northwestern Canada. These were complemented by a P-band PolInSAR survey in 2017 along the same transects. This time series provides a powerful data set with which to evaluate the state of permafrost, active layer thickness, soil moisture, boreal forest structure, above ground biomass, and water surface elevation. Additional studies leverage the PolInSAR data to address fire disturbance and recovery, thermokarst feature development, and retrogressive permafrost thaw megaslumps. Many of these analyses are in progress and will be published separately.

Miller et al. [2023] provides extensive, fully hyperlinked notes on the airborne SAR data. Researchers may discover these data via daily sorties and/or individual flight lines. Alternatively, they may be explored via the interactive map at the JPL UAVSAR data portal, (https://uavsar.jpl.nasa.gov/cgi-bin/data.pl) which provides links to the individual flight line data, maps, and related flight plans that acquired data over one or more of the individual flight lines. We have also identified the ground-based anchor points for each flight line to facilitate comparisons with those data. Calibration and validation data sets as well as many derived products produced by the ABoVE Science Team may be found at the Arctic Boreal Vulnerability Experiment (ABoVE) landing page at the ORNL DAAC (https://daac.ornl.gov/cgi-bin/dataset_lister.pl?p=34).

The example studies (Sections 4 and 5) and multi-instrument synergies (Section 6) described here are only a small portion of the studies currently being undertaken by the ABoVE Science Team and the SAR Working Group. We anticipate many new and innovative uses of the L-band and P-band SAR data as the ABoVE team expands its range of synthesis activities in Phase 3.



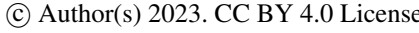 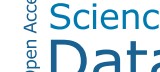


The ABoVE L-band SAR flights planned for 2020 and 2021 were postponed due to the global COVID-
19 pandemic and safety considerations; however, flights were resumed in 2022 and we anticipate at
least one more thaw season campaign in 2024. Finally, we note that the data and analyses discussed
here set the stage for the upcoming NISAR mission (expected launch in 2023). NISAR will deliver
global L- and S-band imagery with a 12-day revisit. Its emphasis on snow- and ice-covered surfaces has
obvious applications in the ABoVE domain, and its global coverage will allow researchers to test the
methods developed for the ABoVE domain across the pan-Arctic. Beyond NISAR, NASA is studying
architectures for the Surface Deformation and Change (SDC) Earth System Observatory Mission and
ESA are developing the Rose-L Copernicus expansion mission. SDC and Rose-L are also envisioned as
an L-band sensors. NISAR, SDC, and Rose-L will all benefit from the ABoVE SAR studies,
algorithmic advances, and lessons learned.
**9 Supplemental Information**
Supplemental Information on the Legacy L-band and P-band flight lines as well as the P-band flight
lines acquired during the ABoVE campaign are provided separately.
**10 Author Contributions**
SJG, CEM, PCG and the ABoVE Science Definition Team developed the preliminary ABoVE
Implementation Plan; this was updated by SJG, CEM, PCG and the ABoVE Science Team. CEM, PCG,
and EH developed the initial flight lines based on the ABoVE Implementation Plan and consultations
with the ABoVE Science and partners. NSP translated the notional flight lines into the UAVSAR
planning system. NSP, YL, MM, PCG, , ELH, and CEM served as Scientist on Board during the data
acquisition flights. YL, SH, BC, NP and the JPL Sub-orbital Radar Science and Engineering Team
(334F) processed the L- and P-band SAR data. CW, KS, MT, JB, LBC, RHC, MM, SW, MM, RJM,
TL, LJ, PS AT, and RD collected cal/val field data. CW and SW coordinated ground cal/val data
acquisitions at the NGEE-Arctic sites. SH, PS, NSP performed the tomoSAR analyses and SH
contributed the text and images for Section 6. KS, MM and the ABoVE SAR Working Group calibrated
the L-band SAR data. MM, AT, RHC processed the P-band data. NSP and YL coordinated the LS-
ASAR flights and the joint BERMS area tomoSAR flights with F-SAR. DS coordinated all data product
submissions to the ORNL DAAC. CEM wrote the initial manuscript and all co-authors contributed to
the final version.
**11 Competing Interests**
The authors declare that they have no conflict of interest





## 12 Acknowledgments

The L- and P-band SAR data acquisitions would not have been possible without the indefatigable support of our NASA pilots and flight crews. We thank John McGrath and the NASA AFRC C-20 (N30502) team as well as Derek Rutovic and the NASA JSC G-III (N995NA) team. We also thank the instrument scientists, operators, and data processing team from the JPL Suborbital Radar Science and Engineering Team (334F) who were essential to the successful execution of these experiments and rapid processing of the resulting data products. The field work supporting the SAR campaigns was made possible by the excellent support from Dan Hodkinson, Sarah Sackett, and the ABoVE Logistics Office. Finally, we thank the data curation team at the Oak Ridge Distributed Active Archive Center for their support and expert advice.

This work was supported by the NASA Terrestrial Ecology Program's Arctic-Boreal Vulnerability Experiment (ABoVE). A portion of this work was performed at the Jet Propulsion Laboratory, California Institute of Technology, under contract with National Aeronautics and Space Administration (80NM0018D0004). Government funding acknowledged.

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

Methods for Detection and Monitoring of Fire Disturbance in the Alaskan Tundra Using a Two-Decade Long
Record of Synthetic Aperture Radar Satellite Images, Remote Sens., 6, 6347–6364,
https://doi.org/10.3390/rs6076347, 2014.





Johnston, C. E., Ewing, S. A., Harden, J. W., Varner, R. K., Wickland, K. P., Koch, J. C., Fuller, C. C., Manies,
K., and Jorgenson, M. T.: Effect of permafrost thaw on CO2 and CH4 exchange in a western Alaska peatland
chronosequence, Environ. Res. Lett., 9, 085004, https://doi.org/10.1088/1748-9326/9/8/085004, 2014.
Jones, B. M., Kolden, C. A., Jandt, R., Abatzoglou, J. T., Urban, F., and Arp, C. D.: Fire Behavior, Weather, and
Burn Severity of the 2007 Anaktuvuk River Tundra Fire, North Slope, Alaska, Arct. Antarct. Alp. Res., 41,
309–316, https://doi.org/10.1657/1938-4246-41.3.309, 2009.
Jones, B. M., Grosse, G., Arp, C. D., Miller, E., Liu, L., Hayes, D. J., and Larsen, C. F.: Recent Arctic tundra fire
initiates widespread thermokarst development, Sci. Rep., 5, 15865, https://doi.org/10.1038/srep15865, 2015.
Jones, M. C., Harden, J., O'Donnell, J., Manies, K., Jorgenson, T., Treat, C., and Ewing, S.: Rapid carbon loss
and slow recovery following permafrost thaw in boreal peatlands, Glob. Change Biol., 23, 1109–1127,
https://doi.org/10.1111/gcb.13403, 2017.
Jorgenson, M. T., Harden, J., Kanevskiy, M., O'Donnell, J., Wickland, K., Ewing, S., Manies, K., Zhuang, Q.,
Shur, Y., Striegl, R., and Koch, J.: Reorganization of vegetation, hydrology and soil carbon after permafrost
degradation across heterogeneous boreal landscapes, Environ. Res. Lett., 8, 035017,
https://doi.org/10.1088/1748-9326/8/3/035017, 2013.
Karion, A., Sweeney, C., Miller, J. B., Andrews, A. E., Commane, R., Dinardo, S., Henderson, J. M., Lindaas, J.,
Lin, J. C., Luus, K. A., Newberger, T., Tans, P., Wofsy, S. C., Wolter, S., and Miller, C. E.: Investigating
Alaskan methane and carbon dioxide fluxes using measurements from the CARVE tower, Atmospheric
Chem. Phys., 16, 5383–5398, https://doi.org/10.5194/acp-16-5383-2016, 2016.
Kobayashi, H., Hiroki, I., and Suzuki, R.: AmeriFlux US-Prr Poker Flat Research Range Black Spruce Forest,
Ver. 3-5, https://doi.org/10.17190/AMF/1246153, 2019.
Kokelj, S. V., Lacelle, D., Lantz, T. C., Tunnicliffe, J., Malone, L., Clark, I. D., and Chin, K. S.: Thawing of
massive ground ice in mega slumps drives increases in stream sediment and solute flux across a range of
watershed scales, J. Geophys. Res. Earth Surf., 118, 681–692, https://doi.org/10.1002/jgrf.20063, 2013.
Kokelj, S. V., Tunnicliffe, J., Lacelle, D., Lantz, T. C., Chin, K. S., and Fraser, R.: Increased precipitation drives
mega slump development and destabilization of ice-rich permafrost terrain, northwestern Canada, Glob.
Planet. Change, 129, 56–68, https://doi.org/10.1016/j.gloplacha.2015.02.008, 2015.
Kokelj, S. V., Lantz, T. C., Tunnicliffe, J., Segal, R., and Lacelle, D.: Climate-driven thaw of permafrost
preserved glacial landscapes, northwestern Canada, Geology, 45, 371–374, https://doi.org/10.1130/G38626.1,
98      2017.
Kyzivat, E. D., Smith, L. C., Pitcher, L. H., Fayne, J. V., Cooley, S. W., Cooper, M. G., Topp, S. N., Langhorst,
T., Harlan, M. E., Horvat, C., Gleason, C. J., and Pavelsky, T. M.: A High-Resolution Airborne Color-
Infrared Camera Water Mask for the NASA ABoVE Campaign, Remote Sens., 11, 2163,
https://doi.org/10.3390/rs11182163, 2019.
Kyzivat, E. D., Smith, L. C., Pitcher, L. H., Fayne, J. V., Cooley, S. W., Cooper, M. G., Topp, S., Langhorst, T.,
Harlan, M. E., Gleason, C. J., and Pavelsky, T. M.: ABoVE: AirSWOT Water Masks from Color-Infrared
Imagery over Alaska and Canada, 2017, ORNL DAAC, https://doi.org/10.3334/ORNLDAAC/1707, 2020.
Lafleur, P. M. and Humphreys, E. R.: Tundra shrub effects on growing season energy and carbon dioxide
exchange, Environ. Res. Lett., 13, 055001, https://doi.org/10.1088/1748-9326/aab863, 2018.
Lantz, T. C. and Turner, K. W.: Changes in lake area in response to thermokarst processes and climate in Old
Crow Flats, Yukon, J. Geophys. Res. Biogeosciences, 120, 513–524, https://doi.org/10.1002/2014JG002744,
10      2015.
Lavalle, M., Hawkins, B., and Hensley, S.: Tomographic imaging with UAVSAR: Current status and new results
from the 2016 AfriSAR campaign, 2017 IEEE International Geoscience and Remote Sensing Symposium
(IGARSS), 2485–2488, https://doi.org/10.1109/IGARSS.2017.8127498, 2017.



Le Toan, T., Quegan, S., Davidson, M. W. J., Balzter, H., Paillou, P., Papathanassiou, K., Plummer, S., Rocca, F., Saatchi, S., Shugart, H., and Ulander, L.: The BIOMASS mission: Mapping global forest biomass to better understand the terrestrial carbon cycle, Remote Sens. Environ., 115, 2850–2860, https://doi.org/10.1016/j.rse.2011.03.020, 2011.

Liljedahl, A., Hinzman, L., Busey, R., and Yoshikawa, K.: Physical short-term changes after a tussock tundra fire, Seward Peninsula, Alaska, J. Geophys. Res. Earth Surf., 112, https://doi.org/10.1029/2006JF000554, 2007.

Lindsay, C., Zhu, J., Miller, A. E., Kirchner, P., and Wilson, T. L.: Deriving Snow Cover Metrics for Alaska from MODIS, Remote Sens., 7, 12961–12985, https://doi.org/10.3390/rs71012961, 2015.

Lipovsky, P. S.: Summary of Yukon Geological Survey permafrost monitoring network results, 2008-2013, in: Yukon Exploration and Geology 2014, edited by: MacFarlane, K. E., Nordling, M. G., and Sack, P. J., Yukon Geological Survey, 113–122, 2015.

Liu, L., Schaefer, K., Zhang, T., and Wahr, J.: Estimating 1992–2000 average active layer thickness on the Alaskan North Slope from remotely sensed surface subsidence, J. Geophys. Res. Earth Surf., 117, https://doi.org/10.1029/2011JF002041, 2012.

Liu, L., Jafarov, E. E., Schaefer, K. M., Jones, B. M., Zebker, H. A., Williams, C. A., Rogan, J., and Zhang, T.: InSAR detects increase in surface subsidence caused by an Arctic tundra fire, Geophys. Res. Lett., 41, 3906–3913, https://doi.org/10.1002/2014GL060533, 2014.

Liu, L., Schaefer, K., Chen, A. C., Gusmeroli, A., Jafarov, E. E., Panda, S. K., Parsekian, A. D., Schaefer, T., Zebker, H. A., and Zhang, T.: Pre-ABoVE: Remotely Sensed Active Layer Thickness, Barrow, Alaska, 2006-2011, ORNL DAAC, https://doi.org/10.3334/ORNLDAAC/1266, 2015a.

Liu, L., Schaefer, K., Chen, A. C., Gusmeroli, A., Jafarov, E. E., Panda, S. K., Parsekian, A. D., Schaefer, T., Zebker, H. A., and Zhang, T.: Pre-ABoVE: Remotely Sensed Active Layer Thickness, Prudhoe Bay, Alaska, 1992-2000, ORNL DAAC, https://doi.org/10.3334/ORNLDAAC/1267, 2015b.

Loboda, T. V., French, N. H. F., Hight-Harf, C., Jenkins, L., and Miller, M. E.: Mapping fire extent and burn severity in Alaskan tussock tundra: An analysis of the spectral response of tundra vegetation to wildland fire, Remote Sens. Environ., 134, 194–209, https://doi.org/10.1016/j.rse.2013.03.003, 2013.

Loboda, T.V., L.K. Jenkins, D. Chen, J. He, and A. Baer. 2022. Burned and Unburned Field Site Data, Noatak, Seward, and North Slope, AK, 2016-2018. ORNL DAAC, Oak Ridge, Tennessee, USA. https://doi.org/10.3334/ORNLDAAC/1919

Lopez-Sanchez, J. M., Ballester-Berman, J. D., Vicente-Guijalba, F., Cloude, S. R., McNairn, H., Shang, J., Skriver, H., Jagdhuber, T., Hajnsek, I., Pottier, E., Marechal, C., Hubert-Moy, L., Corgne, S., Wdowinski, S., Touzi, R., Gosselin, G., Brooks, R., Yamaguchi, Y., and Singh, G.: Agriculture and Wetland Applications, in: Polarimetric Synthetic Aperture Radar: Principles and Application, vol. 25, edited by: Hajnsek, I. and Desnos, Y.-L., Springer International Publishing, Cham, 119–178, https://doi.org/10.1007/978-3-030-56504-6_3, 2021.

Lou, Y., Shimada, J. G., Michel, T. R., Muellerschoen, R. J., Zheng, Y., and Moghaddam, M.: Pre-ABoVE: L1 S-0 Polarimetric Data from AirMOSS P-band SAR, Alaska, 2014-2015, ORNL DAAC, https://doi.org/10.3334/ORNLDAAC/1678, 2019.

Mack, M. C., Bret-Harte, M. S., Hollingsworth, T. N., Jandt, R. R., Schuur, E. A. G., Shaver, G. R., and Verbyla, D. L.: Carbon loss from an unprecedented Arctic tundra wildfire, Nature, 475, 489–492, https://doi.org/10.1038/nature10283, 2011.

Magagi, R., Berg, A. A., Goita, K., Belair, S., Jackson, T. J., Toth, B., Walker, A., McNairn, H., O'Neill, P. E., Moghaddam, M., Gherboudj, I., Colliander, A., Cosh, M. H., Burgin, M., Fisher, J. B., Kim, S.-B., Mladenova, I., Djamai, N., Rousseau, L.-P. B., Belanger, J., Shang, J., and Merzouki, A.: Canadian



Experiment for Soil Moisture in 2010 (CanEx-SM10): Overview and Preliminary Results, IEEE Trans.
Geosci. Remote Sens., 51, 347–363, https://doi.org/10.1109/TGRS.2012.2198920, 2013.
Malone, T., Liang, J., and Packee, E. C.: Cooperative Alaska Forest Inventory, Gen Tech Rep PNW-GTR-785
Portland US Dep. Agric. For. Serv. Pac. Northwest Res. Stn. 42 P, 785, https://doi.org/10.2737/PNW-GTR-
785, 2009.
Marsh, P., Russell, M., Pohl, S., Haywood, H., and Onclin, C.: Changes in thaw lake drainage in the Western
Canadian Arctic from 1950 to 2000, Hydrol. Process., 23, 145–158, https://doi.org/10.1002/hyp.7179, 2009a.
Marsh, P., Lesack, L., Hicks, F., Roberts, A., Hopkinson, C., Solomon, S., Forbes, D., Russell, M., and Haywood,
H.: Hydrology of the Mackenzie Delta: off-channel water storage and delta interaction with the Beaufort Sea,
2009b.
Marsh, P., Bartlett, P., MacKay, M., Pohl, S., and Lantz, T.: Snowmelt energetics at a shrub tundra site in the
western Canadian Arctic, Hydrol. Process., 24, 3603–3620, https://doi.org/10.1002/hyp.7786, 2010.
McCune, B., Arup, U., Breuss, O., Di Meglio, J., Esslinger, T. L., Magain, N., Miadlikowska, J., Miller, A. E.,
Muggia, L., Nelson, P. R., Rosentreter, R., Schultz, M., Sheard, J. W., Tønsberg, T., and Walton, J.:
Biodiversity and ecology of lichens of Katmai and Lake Clark National Parks and Preserves, Alaska,
Mycosphere, 9, 859–930, 2018.
McGuire, A. D., Anderson, L. G., Christensen, T. R., Dallimore, S., Guo, L., Hayes, D. J., Heimann, M.,
Lorenson, T. D., Macdonald, R. W., and Roulet, N.: Sensitivity of the carbon cycle in the Arctic to climate
change, Ecol. Monogr., 79, 523–555, https://doi.org/10.1890/08-2025.1, 2009.
Meddens, A. J. H., Vierling, L. A., Eitel, J. U. H., Jennewein, J. S., White, J. C., and Wulder, M. A.: Developing
5 m resolution canopy height and digital terrain models from WorldView and ArcticDEM data, Remote Sens.
Environ., 218, 174–188, https://doi.org/10.1016/j.rse.2018.09.010, 2018.
Mehra, R. and Ramanujam, V. M.: L S Band Airborne SAR Data Products Calibration, in: 2019 URSI Asia-
Pacific Radio Science Conference (AP-RASC), 2019 URSI Asia-Pacific Radio Science Conference (AP-
RASC), 1–1, https://doi.org/10.23919/URSIAP-RASC.2019.8738681, 2019.
Meyer, G., Humphreys, E. R., Melton, J. R., Cannon, A. J., and Lafleur, P. M.: Simulating shrubs and their
energy and carbon dioxide fluxes in Canada's Low Arctic with the Canadian Land Surface Scheme Including
biogeochemical Cycles (CLASSIC), Biogeosciences Discuss., 1–34, https://doi.org/10.5194/bg-2020-458,
2020.
Michaelian, M., Hogg, E. H., Hall, R. J., and Arsenault, E.: Massive mortality of aspen following severe drought
along the southern edge of the Canadian boreal forest, Glob. Change Biol., 17, 2084–2094,
https://doi.org/10.1111/j.1365-2486.2010.02357.x, 2011.
Michaelides, R. J., Schaefer, K., Zebker, H. A., Parsekian, A., Liu, L., Chen, J., Natali, S., Ludwig, S., and
Schaefer, S. R.: Inference of the impact of wildfire on permafrost and active layer thickness in a
discontinuous permafrost region using the remotely sensed active layer thickness (ReSALT) algorithm,
Environ. Res. Lett., 14, 035007, https://doi.org/10.1088/1748-9326/aaf932, 2019.
Michaelides, R. J., Chen, R. H., Zhao, Y., Schaefer, K., Parsekian, A. D., Sullivan, T., et al. (2021).
Permafrost Dynamics Observatory—part I: Postprocessing and calibration methods of UAVSAR L-
band InSAR data for seasonal subsidence estimation. Earth and Space Science, 8, e2020EA001630.
https://doi.org/10.1029/2020EA001630
Miller, C. E., Griffith, P. C., Goetz, S. J., Hoy, E. E., Pinto, N., McCubbin, I. B., Thorpe, A. K., Hofton, M.,
00        Hodkinson, D., Hansen, C., Woods, J., Larson, E., Kasischke, E. S., and Margolis, H. A.: An overview of
01        ABoVE airborne campaign data acquisitions and science opportunities, Environ. Res. Lett., 14, 080201,
02        https://doi.org/10.1088/1748-9326/ab0d44, 2019.





Miller, C.E., P. Griffith, E.E. Hoy, N. Pinto, Y. Lou, S. Hensley, B. Chapman, J.L. Baltzer, K. Bakian-Dogaheh, W.R. Bolton, L.L. Bourgeau-Chavez, R.H. Chen, B-H. Choe, L.K. Clayton, T.A. Douglas, N.H.F. French, J.E. Holloway, G. Hong, L. Huang, G. Iwahana, L.K. Jenkins, J.S. Kimball, T.V. Loboda, M.C. Mack, P. Marsh, R.J. Michaelides, M. Moghaddam, A.D. Parsekian, K. Schaefer, P. Siqueira, D. Singh, A. Tabatabaeenejad, M.R. Turetsky, R. Touzi, E. Wig, P. Wilson, C.J. Wilson, S.D. Wullschleger, Y. Yi, H.A. Zebker, Y. Zhang, Y. Zhao, and S.J. Goetz. 2023. Summary of the ABoVE L-band and P-band Airborne SAR Surveys. ORNL DAAC, Oak Ridge, Tennessee, USA. https://doi.org/10.3334/ORNLDAAC/2150

Minions, C., Natali, S., Watts, J. D., Ludwig, S., and Risk, D.: ABoVE: Year-Round Soil CO2 Efflux in Alaskan Ecosystems, Version 2, ORNL DAAC, https://doi.org/10.3334/ORNLDAAC/1762, 2020.

Minsley, B. J., Abraham, J. D., Smith, B. D., Cannia, J. C., Voss, C. I., Jorgenson, M. T., Walvoord, M. A., Wylie, B. K., Anderson, L., Ball, L. B., Deszcz-Pan, M., Wellman, T. P., and Ager, T. A.: Airborne electromagnetic imaging of discontinuous permafrost, Geophys. Res. Lett., 39, https://doi.org/10.1029/2011GL050079, 2012.

Moghaddam, M., Tabatabaeenejad, A., Chen, R. H., Saatchi, S. S., Jaruwatanadilok, S., Burgin, M., Duan, X., and Truong-Loi, M. L.: AirMOSS: L2/3 Volumetric Soil Moisture Profiles Derived From Radar, 2012-2015, ORNL DAAC, https://doi.org/10.3334/ORNLDAAC/1418, 2016.

Montesano, P. M., Sun, G., Dubayah, R. O., and Ranson, K. J.: Spaceborne potential for examining taiga–tundra ecotone form and vulnerability, Biogeosciences, 13, 3847–3861, https://doi.org/10.5194/bg-13-3847-2016, 2016.

Myers-Smith, I. H., Forbes, B. C., Wilmking, M., Hallinger, M., Lantz, T., Blok, D., Tape, K. D., Macias-Fauria, M., Sass-Klaassen, U., Lévesque, E., Boudreau, S., Ropars, P., Hermanutz, L., Trant, A., Collier, L. S., Weijers, S., Rozema, J., Rayback, S. A., Schmidt, N. M., Schaepman-Strub, G., Wipf, S., Rixen, C., Ménard, C. B., Venn, S., Goetz, S., Andreu-Hayles, L., Elmendorf, S., Ravolainen, V., Welker, J., Grogan, P., Epstein, H. E., and Hik, D. S.: Shrub expansion in tundra ecosystems: dynamics, impacts and research priorities, Environ. Res. Lett., 6, 045509, https://doi.org/10.1088/1748-9326/6/4/045509, 2011.

Natali, S. M., Schuur, E. A. G., Webb, E. E., Pries, C. E. H., and Crummer, K. G.: Permafrost degradation stimulates carbon loss from experimentally warmed tundra, Ecology, 95, 602–608, https://doi.org/10.1890/13-0602.1, 2014.

National Research Council: Opportunities to Use Remote Sensing in Understanding Permafrost and Related Ecological Characteristics: Report of a Workshop, The National Academies Press, Washington, DC, 2014.

Oechel, W. C., Vourlitis, G. L., Hastings, S. J., Zulueta, R. C., Hinzman, L., and Kane, D.: Acclimation of ecosystem CO 2 exchange in the Alaskan Arctic in response to decadal climate warming, Nature, 406, 978–981, https://doi.org/10.1038/35023137, 2000.

Pan, C. G., Kirchner, P. B., Kimball, J. S., and Du, J.: A Long-Term Passive Microwave Snowoff Record for the Alaska Region 1988–2016, Remote Sens., 12, 153, https://doi.org/10.3390/rs12010153, 2020.

Pavelsky, T. M. and Smith, L. C.: Remote sensing of hydrologic recharge in the Peace-Athabasca Delta, Canada, Geophys. Res. Lett., 35, https://doi.org/10.1029/2008GL033268, 2008.

Pietroniro, A., Peters, D. L., Yang, D., Fiset, J.-M., Saint-Jean, R., Fortin, V., Leconte, R., Bergeron, J., Siles, G. L., Trudel, M., Garnaud, C., Matte, P., Smith, L. C., Gleason, C. J., and Pavelsky, T. M.: Canada's Contributions to the SWOT Mission – Terrestrial Hydrology(SWOT-C TH), Can. J. Remote Sens., 45, 116–138, https://doi.org/10.1080/07038992.2019.1581056, 2019.

Pitcher, L. H., Smith, L. C., Pavelsky, T. M., Fayne, J. V., Cooley, S. W., Altenau, E. H., Moller, D. K., and Arvesen, J.: ABoVE: AirSWOT Radar, Orthomosaic, and Water Masks, Yukon Flats Basin, Alaska, 2015, ORNL DAAC, https://doi.org/10.3334/ORNLDAAC/1655, 2019a.



Pitcher, L. H., Pavelsky, T. M., Smith, L. C., Moller, D. K., Altenau, E. H., Allen, G. H., Lion, C., Butman, D.,
Cooley, S. W., Fayne, J. V., and Bertram, M.: AirSWOT InSAR Mapping of Surface Water Elevations and
Hydraulic Gradients Across the Yukon Flats Basin, Alaska, Water Resour. Res., 55, 937–953,
https://doi.org/10.1029/2018WR023274, 2019b.
Pitcher, L. H., Smith, L. C., Cooley, S. W., Zaino, A., Carlson, R., Pettit, J., Gleason, C. J., Minear, J. T., Fayne,
J. V., Willis, M. J., Hansen, J. S., Easterday, K. J., Harlan, M. E., Langhorst, T., Topp, S. N., Dolan, W.,
Kyzivat, E. D., Pietroniro, A., Marsh, P., Yang, D., Carter, T., Onclin, C., Hosseini, N., Wilcox, E., Moreira,
D., Berge-Nguyen, M., Cretaux, J.-F., and Pavelsky, T. M.: Advancing Field-Based GNSS Surveying for
Validation of Remotely Sensed Water Surface Elevation Products, Front. Earth Sci., 8,
https://doi.org/10.3389/feart.2020.00278, 2020.
Plaza, C., Pegoraro, E., Bracho, R., Celis, G., Crummer, K. G., Hutchings, J. A., Hicks Pries, C. E., Mauritz, M.,
Natali, S. M., Salmon, V. G., Schädel, C., Webb, E. E., and Schuur, E. A. G.: Direct observation of
permafrost degradation and rapid soil carbon loss in tundra, Nat. Geosci., 12, 627–631,
https://doi.org/10.1038/s41561-019-0387-6, 2019.
Porter, C., Morin, P., Howat, I., Noh, M.-J., Bates, B., Peterman, K., Keesey, S., Schlenk, M., Gardiner, J.,
Tomko, K., Willis, M., Kelleher, C., Cloutier, M., Husby, E., Foga, S., Nakamura, H., Platson, M.,
Wethington, M., Williamson, C., Bauer, G., Enos, J., Arnold, G., Kramer, W., Becker, P., Doshi, A.,
D'Souza, C., Cummens, P., Laurier, F., and Bojesen, M.: ArcticDEM,
https://doi.org/10.7910/DVN/OHHUKH, 2018.
Potter, C.: Ecosystem carbon emissions from 2015 forest fires in interior Alaska, Carbon Balance Manag., 13, 2,
https://doi.org/10.1186/s13021-017-0090-0, 2018.
Quegan, S., Le Toan, T., Chave, J., Dall, J., Exbrayat, J.-F., Minh, D. H. T., Lomas, M., D'Alessandro, M. M.,
Paillou, P., Papathanassiou, K., Rocca, F., Saatchi, S., Scipal, K., Shugart, H., Smallman, T. L., Soja, M. J.,
Tebaldini, S., Ulander, L., Villard, L., and Williams, M.: The European Space Agency BIOMASS mission:
Measuring forest above-ground biomass from space, Remote Sens. Environ., 227, 44–60,
https://doi.org/10.1016/j.rse.2019.03.032, 2019.
Quinton, W., Berg, A., Braverman, M., Carpino, O., Chasmer, L., Connon, R., Craig, J., Devoie, É., Hayashi, M.,
Haynes, K., Olefeldt, D., Pietroniro, A., Rezanezhad, F., Schincariol, R., and Sonnentag, O.: A synthesis of
three decades of hydrological research at Scotty Creek, NWT, Canada, Hydrol. Earth Syst. Sci., 23, 2015–
2039, https://doi.org/10.5194/hess-23-2015-2019, 2019.
Quinton, W. L., Adams, J. R., Baltzer, J. L., Berg, A. A., Craig, J. R., and Johnson, E.: Permafrost Ecosystems in
Transition: Understanding and Predicting Hydrological and Ecological Change in the Southern Taiga Plains,
Northeastern British Columbia and Southwestern Northwest Territories, 6, 2015.
Ramanujam, V. M. and Mehra, R.: L S Band SAR Data Processing and Products, in: 2019 URSI Asia-Pacific
Radio Science Conference (AP-RASC), 2019 URSI Asia-Pacific Radio Science Conference (AP-RASC), 1–
1, https://doi.org/10.23919/URSIAP-RASC.2019.8738235, 2019.
Ramanujam, V. M., Suneela, T. J. V. D., and Bhan, R.: ISRO's dual frequency airborne SAR pre-cursor to
NISAR, in: Earth Observing Missions and Sensors: Development, Implementation, and Characterization IV,
Earth Observing Missions and Sensors: Development, Implementation, and Characterization IV, 98810A,
https://doi.org/10.1117/12.2228086, 2016.
Rocha, A. V. and Shaver, G. R.: Burn severity influences postfire CO2 exchange in arctic tundra, Ecol. Appl., 21,
477–489, https://doi.org/10.1890/10-0255.1, 2011a.
Rocha, A. V. and Shaver, G. R.: Postfire energy exchange in arctic tundra: the importance and climatic
implications of burn severity, Glob. Change Biol., 17, 2831–2841, https://doi.org/10.1111/j.1365-
2486.2011.02441.x, 2011b.



Rosen, P.A., Kim, Y., Kumar, R., Misra, T., Bhan, R. and Sagi, V.R., 2017, May. Global persistent SAR
sampling with the NASA-ISRO SAR (NISAR) mission. In *Radar Conference (RadarConf), 2017 IEEE* (pp.
0410-0414). IEEE.
Roy, A., Toose, P., Mavrovic, A., Pappas, C., Royer, A., Derksen, C., Berg, A., Rowlandson, T., El-Amine, M.,
Barr, A., Black, A., Langlois, A., and Sonnentag, O.: L-Band response to freeze/thaw in a boreal forest stand
from ground- and tower-based radiometer observations, Remote Sens. Environ., 237, 111542,
https://doi.org/10.1016/j.rse.2019.111542, 2020.
Saatchi, S. S. and Moghaddam, M.: Estimation of crown and stem water content and biomass of boreal forest
using polarimetric SAR imagery, IEEE Trans. Geosci. Remote Sens., 38, 697–709,
https://doi.org/10.1109/36.841999, 2000.
Saatchi, S., Xu, L., Yang, Y., and Yu, Y.: Evaluation of NISAR Biomass Algorithm in Temperate and Boreal
Forests, IGARSS 2019 - 2019 IEEE International Geoscience and Remote Sensing Symposium, 7363–7366,
https://doi.org/10.1109/IGARSS.2019.8898657, 2019.
Sadeghi, M., Tabatabaeenejad, A., Tuller, M., Moghaddam, M., and Jones, S. B.: Advancing NASA's AirMOSS
P-Band Radar Root Zone Soil Moisture Retrieval Algorithm via Incorporation of Richards' Equation,
Remote Sens., 9, 17, https://doi.org/10.3390/rs9010017, 2017.
Schaefer, K., Chen, A. C., Chen, J., Chen, R. H., Dogaheh, K., Jafarov, E., Liu, L., Michaelides, R. J.,
Moghaddam, M., Parsekian, A. D., Sullivan, T. D., Tabatabaeenejad, A., Thompson, J., and Zebker, H.: The
Permafrost Dynamics Observatory, 2018a.
Schaefer, K., Chen, A. C., Chen, J., Chen, R. H., Dogaheh, K., Jafarov, E., Liu, L., Michaelides, R. J.,
Moghaddam, M., Parsekian, A. D., Sullivan, T. D., Tabatabaeenejad, A., Thompson, J., and Zebker, H.: The
Permafrost Dynamics Observatory (PDO), 2018b.
Schaefer, K., Michaelides, R. J., Chen, R. H., Sullivan, T. D., Parsekian, A. D., Bakian-Dogaheh, K.,
Tabatabaeenejad, A., Moghaddam, M., Chen, J., Chen, A. C., Liu, L., and Zebker, H. A.: ABoVE: Active
Layer Thickness Derived from Airborne L- and P-band SAR, Alaska, 2017, ORNL DAAC,
https://doi.org/10.3334/ORNLDAAC/1676, 2019.
Schaefer, K., R.J. Michaelides, R.H. Chen, T.D. Sullivan, A.D. Parsekian, Y. Zhao, K. Bakian-Dogaheh, A.
Tabatabaeenejad, M. Moghaddam, J. Chen, A.C. Chen, L. Liu, and H.A. Zebker. 2021a. ABoVE: Active
Layer Thickness Derived from Airborne L- and P-band SAR, Alaska, 2017. ORNL DAAC, Oak Ridge,
Tennessee, USA. https://doi.org/10.3334/ORNLDAAC/1796
Schaefer, K., L.K. Clayton, M.J. Battaglia, L.L. Bourgeau-Chavez, R.H. Chen, A.C. Chen, J. Chen, K. Bakian-
Dogaheh, T.A. Douglas, S.E. Grelick, G. Iwahana, E. Jafarov, L. Liu, S. Ludwig, R.J. Michaelides, M.
Moghaddam, S. Natali, S.K. Panda, A.D. Parsekian, A.V. Rocha, S.R. Schaefer, T.D. Sullivan, A.
Tabatabaeenejad, K. Wang, C.J. Wilson, H.A. Zebker, T. Zhang, and Y. Zhao. 2021b. ABoVE: Soil Moisture
and Active Layer Thickness in Alaska and NWT, Canada, 2008-2020. ORNL DAAC, Oak Ridge, Tennessee,
USA. https://doi.org/10.3334/ORNLDAAC/1903
Schuur, E. A. G., Vogel, J. G., Crummer, K. G., Lee, H., Sickman, J. O., and Osterkamp, T. E.: The effect of
permafrost thaw on old carbon release and net carbon exchange from tundra, Nature, 459, 556–559,
https://doi.org/10.1038/nature08031, 2009.
Schuur, T.: AmeriFlux US-EML Eight Mile Lake Permafrost thaw gradient, Healy Alaska (3.5),
https://doi.org/10.17190/AMF/1418678, 2019.
Sellers, P., Hall, F., Margolis, H., Kelly, B., Baldocchi, D., Hartog, G. den, Cihlar, J., Ryan, M. G., Goodison, B.,
Crill, P., Ranson, K. J., Lettenmaier, D., and Wickland, D. E.: The Boreal Ecosystem–Atmosphere Study
(BOREAS): An Overview and Early Results from the 1994 Field Year, Bull. Am. Meteorol. Soc., 76, 1549–
1577, https://doi.org/10.1175/1520-0477(1995)076<1549:TBESAO>2.0.CO;2, 1995.



Sellers, P. J., Hall, F. G., Kelly, R. D., Black, A., Baldocchi, D., Berry, J., Ryan, M., Ranson, K. J., Crill, P. M., Lettenmaier, D. P., Margolis, H., Cihlar, J., Newcomer, J., Fitzjarrald, D., Jarvis, P. G., Gower, S. T., Halliwell, D., Williams, D., Goodison, B., Wickland, D. E., and Guertin, F. E.: BOREAS in 1997: Experiment overview, scientific results, and future directions, J. Geophys. Res. Atmospheres, 102, 28731–28769, https://doi.org/10.1029/97JD03300, 1997.

Serrouya, R., Dickie, M., Lamb, C., van Oort, H., Kelly, A. P., DeMars, C., McLoughlin, P. D., Larter, N. C., Hervieux, D., Ford, A. T., and Boutin, S.: Trophic consequences of terrestrial eutrophication for a threatened ungulate, Proc. R. Soc. B Biol. Sci., 288, 20202811, https://doi.org/10.1098/rspb.2020.2811, 2021.

Sherriff, R. L., Miller, A. E., Muth, K., Schriver, M., and Batzel, R.: Spruce growth responses to warming vary by ecoregion and ecosystem type near the forest-tundra boundary in south-west Alaska, J. Biogeogr., 44, 1457–1468, https://doi.org/10.1111/jbi.12968, 2017.

Shugar, D. H., Clague, J. J., Best, J. L., Schoof, C., Willis, M. J., Copland, L., and Roe, G. H.: River piracy and drainage basin reorganization led by climate-driven glacier retreat, Nat. Geosci., 10, 370–375, https://doi.org/10.1038/ngeo2932, 2017.

Silva, C. A., Duncanson, L., Hancock, S., Neuenschwander, A., Thomas, N., Hofton, M., Fatoyinbo, L., Simard, M., Marshak, C. Z., Armston, J., Lutchke, S., and Dubayah, R.: Fusing simulated GEDI, ICESat-2 and NISAR data for regional aboveground biomass mapping, Remote Sens. Environ., 253, 112234, https://doi.org/10.1016/j.rse.2020.112234, 2021.

Smith, L. C., Pavelsky, T., Lettenmaier, D. P., Gleason, C. J., Pietroniro, A., Applejohn, A., Arvesen, J. C., Bjella, K., Carter, T., Chao, R., Cooley, S. W., Cooper, M. G., Cretaux, J. F., Douglass, T., Faria, D., Fayne, J., Fiset, J. M., Goodman, S., Hanna, B., Harlan, M., Langhorst, T., Marsh, P., Moreira, D. M., Minear, J. T., Onclin, C., Overstreet, B. T., Peters, D., Pettit, J., Pitcher, L. H., Russell, M., Spence, C., Topp, S., Turner, K. W., Vimal, S., Wilcox, E., Woodward, J., Yang, D., and Zaino, A.: AirSWOT flights and field campaigns for the 2017 Arctic-Boreal Vulnerability Experiment (ABoVE), in: AGU Fall Meeting Abstracts, C21F-1176, 2017a.

Smith, S. L., Roy, L.-P., Lewkowicz, A. G., and Chartrand, J.: Ground thermal data collection along the Alaska Highway corridor (KP1559-1895), Yukon, summer 2016, Geological Survey of Canada, 2017b.

Spence, C. and Hedstrom, N.: Hydrometeorological data from Baker Creek Research Watershed, Northwest Territories, Canada, Earth Syst. Sci. Data, 10, 1753–1767, https://doi.org/10.5194/essd-10-1753-2018, 2018.

Sun, G., K. J. Ranson, D. S. Kimes, J. B. Blair, and K. Kovacs (2008), Forest vertical structure from GLAS: An evaluation using LVIS and SRTM data, *Remote Sens. Environ.*, **112**, 107–117, doi:10.1016/j.rse.2006.09.036.

Tabatabaeenejad, A., and M. Moghaddam, 2011. Retrieval of surface and deep soil moisture and effect of moisture profile on inversion accuracy. IEEE Geosci. Remote Sensing Lett., vol. 8, no. 3, pp. 477- 481, May 2011.

Tabatabaeenejad, A., Burgin, M., Duan, X., and Moghaddam, M.: P-Band Radar Retrieval of Subsurface Soil Moisture Profile as a Second-Order Polynomial: First AirMOSS Results, IEEE Trans. Geosci. Remote Sens., 53, 645–658, https://doi.org/10.1109/TGRS.2014.2326839, 2015.

Tabatabaeenejad, A., Chen, R. H., Burgin, M. S., Duan, X., Cuenca, R. H., Cosh, M. H., Scott, R. L., and Moghaddam, M.: Assessment and Validation of AirMOSS P-Band Root-Zone Soil Moisture Products, IEEE Trans. Geosci. Remote Sens., 58, 6181–6196, https://doi.org/10.1109/TGRS.2020.2974976, 2020.

Tank, S., Olefeldt, D., Department of Renewable Resources, University of Alberta, Edmonton, Alberta, Canada, Quinton, W., Centre for Cold Regions and Water Science, Wilfred Laurier University, Waterloo, Ontario, Canada, Spence, C., Environment and Climate Change Canada, Saskatoon, Saskatchewan, Canada, Dion, N., Water Resources Department, Government of Northwest Territories, Yellowknife, Northwest Territories, Canada, Ackley, C., Centre for Cold Regions and Water Science, Wilfred Laurier University, Waterloo, Ontario, Canada, Burd, K., Department of Renewable Resources, University of Alberta, Edmonton, Alberta,





Canada, Hutchins, R., Department of Biological Sciences, University of Alberta, Edmonton, Alberta, Canada,
Mengistu, S., and Department of Biological Sciences, University of Alberta, Edmonton, Alberta, Canada:
Fire in the Arctic: The effect of wildfire across diverse aquatic ecosystems of the Northwest Territories, Polar
Knowl. Aqhaliat Rep., 1, 31–38, https://doi.org/10.35298/pkc.2018.04, 2019.
Touzi, R., Deschamps, A., and Rother, G.: Phase of Target Scattering for Wetland Characterization Using
Polarimetric C-Band SAR, IEEE Trans. Geosci. Remote Sens., 47, 3241–3261,
https://doi.org/10.1109/TGRS.2009.2018626, 2009.
Touzi, R., Omari, K., Sleep, B., and Jiao, X.: Scattered and Received Wave Polarization Optimization for
Enhanced Peatland Classification and Fire Damage Assessment Using Polarimetric PALSAR, IEEE J. Sel.
Top. Appl. Earth Obs. Remote Sens., PP, 1–26, https://doi.org/10.1109/JSTARS.2018.2873740, 2018.
Touzi, R., Hong, G., Motohka, T., Shinichi, S., and De Lisle, D.: Investigation of Compact SAR L and C band
Complementarity for Permafrost Characterization In Arctic Regions, IGARSS 2019 - 2019 IEEE
International Geoscience and Remote Sensing Symposium, 4665–4667,
https://doi.org/10.1109/IGARSS.2019.8898510, 2019a.
Touzi, R., Pawley, S., Hosseini, M., and Jiao, X.: Polarimetric L-band PALSAR2 for Discontinuous Permafrost
Mapping In Peatland Regions, in: IGARSS 2019 - 2019 IEEE International Geoscience and Remote Sensing
Symposium, IGARSS 2019 - 2019 IEEE International Geoscience and Remote Sensing Symposium,
Yokohama, Japan, xvii–ccxxii, https://doi.org/10.1109/IGARSS.2019.8900243, 2019b.
Tsuyuzaki, S., Iwahana, G., and Saito, K.: Tundra fire alters vegetation patterns more than the resultant
thermokarst, Polar Biol., 41, 753–761, https://doi.org/10.1007/s00300-017-2236-7, 2018.
Ullmann, T., Banks, S. N., Schmitt, A., and Jagdhuber, T.: Scattering Characteristics of X-, C- and L-Band
PolSAR Data Examined for the Tundra Environment of the Tuktoyaktuk Peninsula, Canada, Appl. Sci., 7,
595, https://doi.org/10.3390/app7060595, 2017.
Vincent, W. F., Callaghan, T. V., Dahl-Jensen, D., Johansson, M., Kovacs, K. M., Michel, C., Prowse, T., Reist,
J. D., and Sharp, M.: Ecological Implications of Changes in the Arctic Cryosphere, AMBIO, 40, 87–99,
https://doi.org/10.1007/s13280-011-0218-5, 2011.
Walker, B., Wilcox, E. J., and Marsh, P.: Accuracy assessment of late winter snow depth mapping for tundra
environments using Structure-from-Motion photogrammetry1, Arct. Sci., https://doi.org/10.1139/as-2020-
12   0006, 2020.
Walker, X. J., Rogers, B. M., Baltzer, J. L., Cummings, S. R., Day, N. J., Goetz, S. J., Johnstone, J. F., Turetsky,
M. R., and Mack, M. C.: ABoVE: Wildfire Carbon Emissions and Burned Plot Characteristics, NWT, CA,
2014-2016, ORNL DAAC, https://doi.org/10.3334/ORNLDAAC/1561, 2018a.
Walker, X. J., Rogers, B. M., Baltzer, J. L., Cumming, S. G., Day, N. J., Goetz, S. J., Johnstone, J. F., Schuur, E.
A. G., Turetsky, M. R., and Mack, M. C.: Cross-scale controls on carbon emissions from boreal forest
megafires, Glob. Change Biol., 24, 4251–4265, https://doi.org/10.1111/gcb.14287, 2018b.
Walker, X. J., Baltzer, J. L., Laurier, W., Cumming, S. G., Day, N. J., Goetz, S. J., Johnstone, J. F., Potter, S.,
Rogers, B. M., Schuur, E. a. G., Turetsky, M. R., and Mack, M. C.: ABoVE: Characterization of Carbon
Dynamics in Burned Forest Plots, NWT, Canada, 2014, ORNL DAAC,
https://doi.org/10.3334/ORNLDAAC/1664, 2019a.
Walker, X. J., Baltzer, J. L., Cumming, S. G., Day, N. J., Ebert, C., Goetz, S., Johnstone, J. F., Potter, S., Rogers,
B. M., Schuur, E. A. G., Turetsky, M. R., and Mack, M. C.: Increasing wildfires threaten historic carbon sink
of boreal forest soils, Nature, 572, 520–523, https://doi.org/10.1038/s41586-019-1474-y, 2019b.
Whalen, D., Forbes, D. L., Hopkinson, C., Lavergne, J. C., Manson, G. K., Marsh, P., and Solomon, S. M.:
Topographic LiDAR-Providing a new perspective in the Mackenzie Delta, 2009.



Whitley, M., Frost, G. V., Jorgenson, M. T., Macander, M., Maio, C. V., and Winder, S. G.: ABoVE: Permafrost
Measurements and Distribution Across the Y-K Delta, Alaska, 2016, ORNL DAAC,
https://doi.org/10.3334/ORNLDAAC/1598, 2018a.
Whitley, M. A., Frost, G. V., Jorgenson, M. T., Macander, M. J., Maio, C. V., and Winder, S. G.: Assessment of
LiDAR and Spectral Techniques for High-Resolution Mapping of Sporadic Permafrost on the Yukon-
Kuskokwim Delta, Alaska, Remote Sens., 10, 258, https://doi.org/10.3390/rs10020258, 2018b.
Wilcox, E. J., Keim, D., Jong, T. de, Walker, B., Sonnentag, O., Sniderhan, A. E., Mann, P., and Marsh, P.:
Tundra shrub expansion may amplify permafrost thaw by advancing snowmelt timing, Arct. Sci.,
https://doi.org/10.1139/as-2018-0028, 2019.
Wilson, C., Dann, J., Bolton, R., Charsley-Groffman, L., Jafarov, E., Musa, D., and Wullschleger, S.: In Situ Soil
Moisture and Thaw Depth Measurements Coincident with Airborne SAR Data Collections, Barrow and
Seward Peninsulas, Alaska, 2017, https://doi.org/10.5440/1423892, 2021.
Woodward, A. and Beever, E. A.: Conceptual ecological models to support detection of ecological change on
Alaska National Wildlife Refuges, 2011.
Yi, Y., Kimball, J. S., Chen, R. H., Moghaddam, M., Reichle, R. H., Mishra, U., Zona, D., and Oechel, W. C.:
Characterizing permafrost active layer dynamics and sensitivity to landscape spatial heterogeneity in Alaska,
The Cryosphere, 12, 145–161, https://doi.org/10.5194/tc-12-145-2018, 2018.
Zhang, Y., Touzi, R., Feng, W., Hong, G., Lantz, T. C., and Kokelj, S. V.: Landscape-scale variations in near-
surface soil temperature and active-layer thickness: Implications for high-resolution permafrost mapping,
Permafr. Periglac. Process. J., 2021. https://doi.org/10.1002/ppp.2104
Zhao, Y., Chen, R.H., Bakina-Dogaheh, K., Whitcomb, J., Yi, Y., Kimball, J. S., Moghaddam, M.: Mapping
Boreal Forest Species and Canopy Height using Airborne SAR and Lidar Data in Interior Alaska, IGARSS
2022 - 2022 IEEE International Geoscience and Remote Sensing Symposium, Kuala Lumpur, Malaysia,
2022, pp. 4955-4958, doi: 10.1109/IGARSS46834.2022.9883311.