# Peer review of "The ABoVE L-band and P-band Airborne SAR Surveys 1"

_Earth System Science Data, 2021_

## Author Response (AR1)

Reviewer Comments

- **RC1**: 'Comment on essd-2021-172', Anonymous Referee #1, 07 Nov 2023

This is a very interesting and unique data set, very relevant in the field of permafrost analyis with SAR data. The paper is enjoyable to read and complete. It gives an excellent background about the sensor and data acquistion. The data presentation is well organised and uses various links where SAR data and derived products can be found. Very nice is the inclusion of a chapter about synergies with other sensors, which well completes the presented ABoVE initiative.

We thank Reviewer 1 for a thorough reading of the manuscript.

I found only two minor issues:

- Page 18, line 57: The second "and" seems to be wrong and should be deleted
Response: Correction Made. Thank you for catching this error. Line 556 in Word file

- Page 20, line 20/21: The usage of the term "Random Forest" in the context of forest analysis is quite confusing if you are not aware the it is a machine learning technique. And unskilled reader will not understand what you mean. Please reformulate this sentence to point out that you mean a ML-technique and not a real forest here.

Response: text changed to
They employed machine learning (random forests) to train separate regression models for canopy height mapping and a classification model for forest species mapping.

RC2: 'Comment on essd-2021-172', Anonymous Referee #2, 22 Jan 2024

Miller et al. present an annotated guide to the L-band and P-band airborne SAR data acquired during the 2017, 2018, 2019 and 2022 Arctic-Boreal Vulnerability Experiment ABoVE airborne campaigns across Alaska and northwestern Canada.

Airborne data are a key component of NASA's ABoVE. The airborne SAR data presented here are highly useful for a wide range of applications for permafrost and land surface monitoring and research. In addition to the presented core data between 2017 to 2022,

also legacy acquisitions between 2012 and 2015 were included for completeness and to enable longer times series creation.

This paper is well written with clear motivations, a good structure, and extensive overview maps. The authors provide a detailed overview on the ~80 SAR flight lines and how each fits into the ABoVE experimental design. In the additional data report, tables, and hyperlinks give direct access to every flight plan as well as individual flight lines. The data sets are well-published, it is easy to navigate and download data. The data sets are well-prepared and described by experienced experts with high quality.

Beyond on providing information on the airborne data the authors also provide information on -the involvement of the Northerners in planning and during airborne campaigns, -more detailed application in some case studies, e.g. permafrost active layer and soil moisture mapping, thaw slumps, vegetation structure and biomass, -legacy airborne campaigns, -external airborne campaigns in their domain, -multi-instrument synergies, -value as preparation for upcoming space missions.

We thank Reviewer 2 for carefully reading of the manuscript and pointing out several areas requiring clarification.

There are only minor issues

- **In general**: 'et al.' is missing in the citations

 Response: "et al." added to references throughout

- **Abstract** L53 Boreal Ecosystem Research and Monitoring Sites (BERMS) in northern Saskatchewan, CA

Response: Text changed to include full call out for "Boreal Ecosystem Research and Monitoring Sites (BERMS)"

- **chapter 9 Supplemental Information**
- What is the difference between the supplemental data (chapter 9) and the data linked to Miller et al. 2023?
- In chapter 9  It reads: "Supplemental Information on the Legacy L-band and P-band flight lines as well as the P-band flight lines acquired during the ABoVE campaign are provided separately."

- What is the difference to the data in the detailed report Miller et al. 2023? - They also seem to contain the legacy flight lines and the P-band flight lines?
- "…This dataset contains tables containing Airborne flight metadata from synthetic aperture radar (SAR) surveys from 2012 to 2022 in Alaska and Canada. NASA's Arctic Boreal Vulnerability Experiment (ABoVE) conducted airborne SAR surveys of over 120,000 km2 in Alaska and northwestern Canada during 2017, 2018, 2019, and 2022. Legacy lines acquired between 2012 and 2015 by other projects are included for completeness and to enable longer times series creation. The data files and companion file contain L-band and P-band airborne SAR metadata acquired during the ABoVE airborne campaigns…"

Response: Thank you for identifying this issue. We have clarified the language in Section 9 to reflect that the SI is one of the files that accompanies the data download from Miller et al. [2023]:
"The Supplemental Information (SI) contains detailed descriptions of all L-band flight lines plus tables with hyperlinks to all L-band lines and sorties. Additionally, the SI includes tables and links to the P-band flight lines acquired during the ABoVE campaigns and to all Legacy L-band and P-band flight lines. The SI is identical to the file  <Summary ABoVE L- & P-Band SAR Surveys - hyperlinked.pdf> that may be found in the /data folder of the uncompressed data download from Miller et al. [2023; https://doi.org/10.3334/ORNLDAAC/2150]." Lines 684-689